# GROUP-ROBUST SAMPLE REWEIGHTING FOR SUBPOPULATION SHIFTS VIA INFLUENCE FUNCTIONS

**Rui Qiao**[1 2 †]  **Zhaoxuan Wu**[2]  **Jingtan Wang**[1 3]  **Pang Wei Koh**[4]  **Bryan Kian Hsiang Low**[1 2]
[1]National University of Singapore    [2]Singapore-MIT Alliance for Research and Technology
[3]Agency for Science, Technology and Research (A*STAR)    [4]University of Washington
{rui.qiao,jingtan.w}@u.nus.edu    zhaoxuan.wu@smart.mit.edu
pangwei@cs.washington.edu    lowkh@comp.nus.edu.sg

## ABSTRACT

Machine learning models often have uneven performance among subpopulations (a.k.a., groups) in the data distributions. This poses a significant challenge for the models to generalize when the proportions of the groups shift during deployment. To improve robustness to such shifts, existing approaches have developed strategies that train models or perform hyperparameter tuning using the group-labeled data to minimize the worst-case loss over groups. However, a non-trivial amount of high-quality labels is often required to obtain noticeable improvements. Given the costliness of the labels, we propose to adopt a different paradigm to enhance group label efficiency: utilizing the group-labeled data as a target set to optimize the weights of other group-unlabeled data. We introduce Group-robust Sample Reweighting (GSR), a two-stage approach that first learns the representations from group-unlabeled data, and then tinkers the model by iteratively retraining its last layer on the reweighted data using influence functions. Our GSR is theoretically sound, practically lightweight, and effective in improving the robustness to subpopulation shifts. In particular, GSR outperforms the previous state-of-the-art approaches that require the same amount or even more group labels. Our code is available at https://github.com/qiaoruiyt/GSR.

## 1 INTRODUCTION

Subpopulation shift refers to the change in the proportion of groups (as defined by the class, attributes, or both) in data distribution between the training and deployment phases (Koh et al., 2021; Yang et al., 2023). Due to the nature of subpopulations or the selection bias (Dwork et al., 2012; Zadrozny, 2004), the training data is often composed of majority and minority groups that are overrepresented and underrepresented, respectively. Common examples include photos of waterbirds frequently being taken with a water background instead of a land background (Sagawa et al., 2019) and people are more likely to be holding a vacuum than fixing it (Nan et al., 2021). Such group imbalance can cause some features from the majority group to be spuriously correlated with the labels, leading to possible undesirable shortcuts in training deep neural networks (Geirhos et al., 2020; Hermann et al., 2024; Shah et al., 2020). As a result, standard training with empirical risk minimization (ERM) often overly relies on such shortcuts and fails to attend to the minority groups, leading to worsened model generalization under subpopulation shifts, especially when the proportion of the minority groups increases (Arjovsky et al., 2019; Nagarajan et al., 2020; Sagawa et al., 2020). Thus, it is important to develop algorithms that are group-robust to subpopulation shifts, experiencing minimal performance degradation as measured by the worst-group accuracy.

The group labels can be used to construct balanced groups or minimize the worst-group loss during training (Idrissi et al., 2022; Sagawa et al., 2019). These techniques often lead to improved group robustness compared to ERM. However, group labels can be expensive to obtain, necessitating approaches that maximize the gain in group robustness from limited data. Given a group-labeled dataset $\mathcal{D}^{\mathrm{L}}$, the two mainstream categories are either using $\mathcal{D}^{\mathrm{L}}$ for model selection (Liu et al., 2021;

---

† Partial work done while the author was a visiting scholar at the University of Washington.

Zhang et al., 2022), or using $\mathcal{D}^{\mathrm{L}}$ to directly train the model parameters (Kirichenko et al., 2022). However, these strategies may not always be the best way to exploit the group information, since they either underutilize the group labels, or restrict the most crucial part of the training to only $\mathcal{D}^{\mathrm{L}}$ while neglecting the potential of the more accessible data without group labels. Therefore, we advocate an in-between training paradigm: instead of using $\mathcal{D}^{\mathrm{L}}$ to train the model parameters directly, we use them to iteratively optimize the weights of a group-unlabeled dataset $\mathcal{D}^{\mathrm{U}}$, which are more realistic to obtain. Then, the model parameters are trained using a weighted ERM objective on these data.

Optimizing the sample weights is an established idea that necessitates solving a bilevel optimization problem (Ren et al., 2018). The inner loop of the bilevel optimization performs standard model training on a weighted objective, while the outer loop optimizes the sample weights. The two main challenges hindering its wider adoption include: (1) the unavoidable trade-off between computational cost and accuracy, and (2) the effectiveness when the model is overparameterized. For (1), calculating the outer loop gradient requires backpropagating through the entire training process, which is computationally prohibitive for deep neural networks. Conventional approaches use a one-step truncated backpropagation that relies only on the last-step gradient of the model training for computational feasibility (Shaban et al., 2019; Zhou et al., 2022). However, this approximation can be imprecise as it fails to account for the curvature of the loss landscape and the training dynamics of the model. For (2), training a weighted objective with overparameterized neural networks is likely to converge to the same solution as with ERM (Sagawa et al., 2019; Zhai et al., 2022). Regularization techniques need to be applied such that training with weighted objectives leads to meaningfully different solutions than training with ERM (Byrd & Lipton, 2019; Sagawa et al., 2019; Zhai et al., 2022).

To address the two challenges, we propose a two-stage method named Group-robust Sample Reweighting (GSR), which iteratively reweights individual training samples to improve group robustness. We utilize last-layer retraining (LLR), a lightweight method that retrains the last linear layer of neural network (Kang et al., 2020; Kirichenko et al., 2022). LLR simplifies the inner loop of the bilevel optimization into a convex optimization problem. Importantly, it facilitates our application of the influence function, a technique that can be derived from implicit differentiation (Koh & Liang, 2017; Krantz & Parks, 2002), to utilize the Hessian to accurately estimate the gradient of sample weight updates without backpropagating through the entire training trajectory. In contrast to methods that rely on one-step truncated backpropagation approximations (Zhou et al., 2022), our approach leverages the fact that LLR is both inexpensive even with Hessian computation and effective for enhancing group robustness. To perform GSR, we split the group-unlabeled dataset $\mathcal{D}^{\mathrm{U}}$ into a held-out set $\mathcal{D}^{\mathrm{U\text{-}h}}$ and a remaining set $\mathcal{D}^{\mathrm{U\text{-}r}}$. The first stage performs unweighted representation learning on $\mathcal{D}^{\mathrm{U\text{-}r}}$. The second stage utilizes the influence function to iteratively optimize the weights of $\mathcal{D}^{\mathrm{U\text{-}h}}$ for the worst-group loss in $\mathcal{D}^{\mathrm{L}}$ achieved through LLR. Our method outperforms the state-of-the-art method that uses the *same* amount of group labels and even outperforms methods that require *more* group labels.

The specific contributions of this work include the following:

- We propose to better leverage the high-quality group-labeled data for group robustness with an alternative paradigm by using them to reweight other samples instead of directly training on them.
- We devise an efficient strategy based on implicit differentiation for group-robust sample reweighting, which becomes accurate and computationally feasible via the synergy with last-layer retraining.
- We empirically demonstrate the performance advantages of our lightweight approach on improving the group robustness for both vision and natural language tasks.

## 2 PROBLEM FORMULATION AND PRELIMINARIES

Let $(\mathcal{X}, \mathcal{Y}, \mathcal{G})$ denote the space of input, class label, and groups of subpopulation respectively. Denote a dataset by $\mathcal{D} = \{z_1, \ldots, z_n\}$, where each data point $z_i := (x_i, y_i, g_i) \in \mathcal{X} \times \mathcal{Y} \times \mathcal{G}$ is sampled i.i.d. from a data distribution $\mathcal{P}$. For $g \in \mathcal{G}$, define $\mathcal{D}_g := \{z_i \in \mathcal{D} | g_i = g\}$ as the subset of data from group $g$, sampled from the group data distribution $\mathcal{Q}_g$. We assume that $\mathcal{P}$ can be decomposed into $\mathcal{P} = \sum_{g \in \mathcal{G}} \pi_g \mathcal{Q}_g$ where the mixing ratio $\pi_g \in [0, 1]$, $\sum_{g \in \mathcal{G}} \pi_g = 1$. As a result, any subpopulation shifts can be represented by adjusting $\pi_g$. Depending on the group-label availability, datasets can be categorized as $\mathcal{D}^{\mathrm{L}}, \mathcal{D}^{\mathrm{U}}$ for group-labeled, group-unlabeled datasets. Meanwhile, depending on the functionality, datasets can be categorized as $\mathcal{D}^{\mathrm{tr}}, \mathcal{D}^{\mathrm{v}}, \mathcal{D}^{\mathrm{tar}}$, denoting the training, validation, target sets, respectively. The target set $\mathcal{D}^{\mathrm{tar}}$ guides the sample weight updates and we reserve the notion of

validation set $\mathcal{D}^{\mathrm{v}}$ only for hyperparameter and model selection. Let $N = \{1, \ldots, n\}$ be the indices of the training set. We use d for the total derivative and use $\partial$ or $\nabla$ for the partial derivative.

ERM optimizes the model parameters $\theta$ with $\hat{\mathcal{R}}(\mathcal{D}^{\mathrm{tr}}; \theta) = \sum_{i \in N} \frac{1}{n} \ell(x_i, y_i; \theta)$ where $\ell$ is the loss function. Let $w \in \mathbb{R}^n$ be the weight vector for all training samples. A weighted training objective assigns different importance to the training data points, $\hat{\mathcal{R}}(\mathcal{D}^{\mathrm{tr}}; w, \theta) = \sum_{i \in N} w_i \ell(x_i, y_i; \theta)$. Since the change of $\pi$ is not known at test time, we evaluate the robustness to distribution shifts of a model with parameters $\theta$ using the worst-group risk (Sagawa et al., 2019):

$$\hat{\mathcal{R}}^{\mathrm{WG}}(\mathcal{D}; \theta) = \max_{g \in \mathcal{G}} \hat{\mathcal{R}}(\mathcal{D}_g; \theta) .$$

To minimize the worst-group risk, existing methods (Kirichenko et al., 2022; Sagawa et al., 2019) have focused on directly training on group-annotated dataset $\mathcal{D}^{\mathrm{L}}$ using the objective $\hat{\mathcal{R}}^{\mathrm{WG}}(\mathcal{D}^{\mathrm{L}}; \theta)$. However, they require having a sufficient amount of data for each group. Alternatively, we can assume the availability of some group-unlabeled data $\mathcal{D}^{\mathrm{U}}$ (e.g., standard training set). By viewing the sample weights of $\mathcal{D}^{\mathrm{U}}$ as another set of parameters and optimizing it jointly with $\theta$ w.r.t. the target set $\mathcal{D}^{\mathrm{tar}} \subseteq \mathcal{D}^{\mathrm{L}}$, we arrive at a bilevel formulation of the minimax problem (Zhou et al., 2022):

$$\min_{w \in S} \max_{g \in \mathcal{G}} \hat{\mathcal{R}}(\mathcal{D}_g^{\mathrm{tar}}; \hat{\theta}^*)$$
$$\text{s.t. } \hat{\theta}^* = \arg\min_{\theta} \hat{\mathcal{R}}(\mathcal{D}^{\mathrm{tr}}; w, \theta) , \tag{1}$$

where $S = \{w = [w_1, w_2, \cdots, w_n] \in \mathbb{R}^n | w_i \geq 0 \, \forall i \in N$, and $\sum_{i \in N} w_i = 1\}$. The inner loop performs standard model training on the weighted objective with variants of gradient descent. To optimize the sample weights of the training set in the outer loop, we can similarly perform gradient descent using its total derivative:

$$\frac{\mathrm{d}\hat{\mathcal{R}}(\mathcal{D}^{\mathrm{tar}}; \hat{\theta}^*)}{\mathrm{d}w} = \nabla_\theta \hat{\mathcal{R}}(\mathcal{D}^{\mathrm{tar}}; \hat{\theta}^*) \frac{\mathrm{d}\hat{\theta}^*}{\mathrm{d}w} .$$

Since $\hat{\theta}^*$ is typically obtained via gradient descent, calculating $\frac{\mathrm{d}\hat{\theta}^*}{\mathrm{d}w}$ involves unrolling the entire training trajectory and backpropagating through it, which is overly expensive to compute especially when the model is an overparameterized deep neural network. To efficiently approximate $\frac{\mathrm{d}\hat{\theta}^*}{\mathrm{d}w}$, MAPLE (Zhou et al., 2022) adopts one-step truncated backpropagation (Shaban et al., 2019):

$$\frac{\mathrm{d}\hat{\theta}^*}{\mathrm{d}w} \overset{\text{one-step}}{\approx} \nabla_w \theta_T = -\eta \frac{\partial^2 \hat{\mathcal{R}}(\mathcal{D}^{\mathrm{tr}}; w, \theta_{T-1})}{\partial\theta\partial w} , \tag{2}$$

where $\eta$ is the learning rate. Subsequently, $w$ is updated with projected gradient descent. This approximation essentially relies on the last-step gradient w.r.t. each training instance $z_i$, i.e., $\nabla_{w_i} \theta_T = -\eta \nabla_\theta \hat{\mathcal{R}}(z_i; \theta_{T-1}) \approx -\eta \nabla_\theta \hat{\mathcal{R}}(z_i; \theta_T)$, derived in Appendix A.2. After dropping $\eta$, we can rewrite:

$$\frac{\mathrm{d}\hat{\mathcal{R}}(\mathcal{D}^{\mathrm{tar}}; \hat{\theta}^*)}{\mathrm{d}w_i} \overset{\text{MAPLE}}{\approx} -\nabla_\theta \hat{\mathcal{R}}(\mathcal{D}^{\mathrm{tar}}; \theta_T)^\top \nabla_\theta \hat{\mathcal{R}}(z_i; \theta_T) . \tag{3}$$

To interpret, MAPLE essentially updates sample weights according to the inner product between the gradient of the target loss and the gradient of the unweighted training loss for each sample upon convergence. Performing bilevel optimization with this approximation is reasonable because it tends to upweight training points that share similar gradient directions with the target sets. However, it is also imprecise, as it neither accounts for the curvature of the loss landscape w.r.t. the current model nor fully captures the training dynamics, where $w_i$ could significantly affect the batch gradient and optimization trajectory. Reweighting data points based solely on last-step gradients is shortsighted and can lead to suboptimal convergence.

## 3 Group-robust Sample Reweighting via Implicit Differentiation

Our goal is to develop a technique that updates the sample weights more accurately and efficiently to optimize the bilevel minimax objective in Equation 1. In this section, we first establish the connection between reweighting samples via implicit differentiation and the influence function (Koh & Liang, 2017) from the perspective of bilevel optimization. Then, we present an effective integration of the

influence function and adaptive aggregation (Sagawa et al., 2019) over the groups to optimize the minimax objective in the outer loop.

Implicit differentiation can be used to calculate the gradient w.r.t. the sample weights without backpropagating through the inner loop of the objective according to the implicit function theorem (IFT) (Krantz & Parks, 2002). Assuming an unconstrained and strictly convex inner objective, it implies that (1) the gradient w.r.t. the model parameters $\theta$ diminishes to zero upon convergence to the inner loop optimum, i.e., $\nabla_\theta \hat{\mathcal{R}}(\mathcal{D}^{\text{tr}}; w, \hat{\theta}^*) = 0$, and (2) the Hessian of the weighted training objective evaluated at $\theta = \hat{\theta}^*$, $H_{\hat{\theta}^*, w} := \nabla_\theta^2 \hat{\mathcal{R}}(\mathcal{D}^{\text{tr}}; w, \hat{\theta}^*)$, is invertible (Boyd & Vandenberghe, 2004). This satisfies two of the conditions required to apply the IFT. In addition, IFT further requires the inner objective to be twice continuously differentiable. We formally state the assumptions below.

**Assumption 3.1.** The inner objective $\hat{\mathcal{R}}(\mathcal{D}^{\text{tr}}; w, \theta)$ is unconstrained, strictly convex, and twice continuously differentiable with respect to $\theta$, $\forall w \in S$ (Equation 1).

Although appearing as restrictive, these assumptions can be easily satisfied through last-layer retraining, as detailed in Section 4. Under Assumption 3.1, as $\nabla_\theta \hat{\mathcal{R}}(\mathcal{D}^{\text{tr}}; w, \hat{\theta}^*) = 0$, the outer gradient $\frac{d\hat{\theta}^*}{dw} = \begin{bmatrix} \frac{d\hat{\theta}^*}{dw_1} & \frac{d\hat{\theta}^*}{dw_2} & \cdots & \frac{d\hat{\theta}^*}{dw_n} \end{bmatrix}^\top \in \mathbb{R}^{n \times 1}$ can be calculated *exactly* via implicit differentiation as:

$$\frac{d\hat{\theta}^*}{dw} = -H_{\hat{\theta}^*, w}^{-1} \nabla_w \nabla_\theta \hat{\mathcal{R}}(\mathcal{D}^{\text{tr}}; w, \hat{\theta}^*) , \tag{4}$$

$$\frac{d\hat{\theta}^*}{dw_i} = -H_{\hat{\theta}^*, w}^{-1} \nabla_\theta \hat{\mathcal{R}}(z_i; \hat{\theta}^*) = -H_{\hat{\theta}^*, w}^{-1} \nabla_\theta \ell(z_i; \hat{\theta}^*) . \tag{5}$$

The derivation is in Appendix A.3. To highlight, Equation 5 uses the first-order gradient of the *unweighted* ERM loss which is independent of $w_i$. Hence, the magnitude of $w_i$ does *not* directly affect the scale of its gradient, even if $|w_i| \to 0$. Instead, $w_i$ *indirectly* affects the gradients through the trained model parameters $\hat{\theta}^*$ and the Hessian $H_{\hat{\theta}^*, w}$. The total derivative of $\hat{\mathcal{R}}$ w.r.t. $w_i$ is thus:

$$\frac{d\hat{\mathcal{R}}(\mathcal{D}^{\text{tar}}; \hat{\theta}^*)}{dw_i} = -\nabla_\theta \hat{\mathcal{R}}(\mathcal{D}^{\text{tar}}; \hat{\theta}^*)^\top H_{\hat{\theta}^*, w}^{-1} \nabla_\theta \hat{\mathcal{R}}(z_i; \hat{\theta}^*) . \tag{6}$$

In comparison to Equation 3, the key difference lies in the presence of the inverse Hessian term between the two gradients. The Hessian $H_{\hat{\theta}^*, w}$ describes the loss landscape of $\hat{\theta}^*$ w.r.t. the training dataset $\mathcal{D}^{\text{tr}}$ and their sample weights $w$. A more detailed discussion on the role of Hessian is available in Appendix C. We will show later in Section 5 that without the Hessian, the last-step approximated gradient (Equation 3) of the outer objective function is inaccurate, often leading to suboptimal results.

The derived form in Equation 6 via implicit differentiation exactly matches the influence function, which estimates the change in model parameters if a training data point $z_i$ is upweighted infinitesimally (Cook & Weisberg, 1982; Koh & Liang, 2017). Since the influence function is well-defined for weighted objectives, we slightly generalize the notation by incorporating $w$ into the expressions. Given a model $\hat{\theta}^*$ trained on weighted samples, the influence of upweighting $z_i \in \mathcal{D}^{\text{tr}}$ on the loss of model predicting a target instance $z' \in \mathcal{D}^{\text{tar}}$ (or a target set $\mathcal{D}^{\text{tar}}$) is

$$\mathcal{I}(z_i, z'; \hat{\theta}^*, w) := -\nabla_\theta \hat{\mathcal{R}}(z'; \hat{\theta}^*)^\top H_{\hat{\theta}^*, w}^{-1} \nabla_\theta \hat{\mathcal{R}}(z_i; \hat{\theta}^*) . \tag{7}$$

Thus, we can use the influence function to represent the gradient derived from implicit differentiation. For completeness, we define the *per-sample* influence of a dataset to be $\widetilde{\mathcal{I}}(\mathcal{D}^{\text{tr}}, z'; \hat{\theta}^*, w) := \begin{bmatrix} \mathcal{I}(z_1, z'; \hat{\theta}^*, w) & \mathcal{I}(z_2, z'; \hat{\theta}^*, w) & \cdots & \mathcal{I}(z_n, z'; \hat{\theta}^*, w) \end{bmatrix}^\top \in \mathbb{R}^{n \times 1}$. Then, we can calculate the sample weight update based on the per-sample influence in $\mathcal{D}^{\text{tr}}$ on each group $g \in \mathcal{G}$ in $\mathcal{D}^{\text{tar}}$:

$$\widetilde{\mathcal{I}}(\mathcal{D}^{\text{tr}}, \mathcal{D}_g^{\text{tar}}; \hat{\theta}^*, w) = \frac{1}{|\mathcal{D}_g^{\text{tar}}|} \sum_{z' \in \mathcal{D}_g^{\text{tar}}} \widetilde{\mathcal{I}}(\mathcal{D}^{\text{tr}}, z'; \hat{\theta}^*, w) . \tag{8}$$

To optimize the minimax objective in the outer loop, the gradient of the worst-group risk w.r.t. the sample weights can be calculated via its corresponding influence scores:

$$\frac{d\hat{\mathcal{R}}^{\text{WG}}(\mathcal{D}^{\text{tar}}; \hat{\theta}^*)}{dw} = \widetilde{\mathcal{I}}(\mathcal{D}^{\text{tr}}, \mathcal{D}_{g^*}^{\text{tar}}; \hat{\theta}^*, w), \quad g^* = \arg\max_{g \in \mathcal{G}} \hat{\mathcal{R}}(\mathcal{D}_g^{\text{tar}}; \hat{\theta}^*) . \tag{9}$$

However, only optimizing for the worst group in one step according to Equation 9 can be inefficient, especially when multiple groups have high error rates. To obtain a more efficient and smoother optimization trajectory, we utilize an adaptive aggregation (Sagawa et al., 2019) of the influence scores across groups. The aggregation weights are updated multiplicatively based on group error rates to better optimize the sample weights, illustrated in lines 7-14 of Algorithm 1.

# 4 Exact Group-robust Sample Reweighting with Last-layer Retraining

In this section, we introduce the Group-robust Sample Reweighting with last-layer retraining (GSR) algorithm. The algorithm is built on the observation that last-layer retraining (LLR) (Kang et al., 2020; Kirichenko et al., 2022) can lead to a strongly convex (a subset of strictly convex) inner objective that satisfies the assumptions required for implicit differentiation. Next, we delineate the benefits of LLR and its integration into our GSR algorithm.

## 4.1 Synergy with Last-layer Retraining

LLR is a lightweight method designed to mitigate spurious correlations and improve group robustness (Kirichenko et al., 2022; LaBonte et al., 2024; Qiu et al., 2023). In essence, it performs deep representation learning with ERM, followed by retraining only the last layer on a separate group-balanced dataset while freezing the remaining model parameters. Empirical evidence has shown that the initial ERM model has learned sufficient representations for all groups, including minority groups, and over-reliance on spurious correlations can be significantly reduced by simply fine-tuning the last linear classification layer of the model (Izmailov et al., 2022; Rosenfeld et al., 2022). As a result, despite being lightweight, last-layer retraining while freezing the ERM-learned representation has become the state-of-the-art approach for improving group robustness across various settings.

Notably, LLR synergies well with our proposed sample reweighting via implicit differentiation (Section 3), offering three significant benefits. Firstly, employing LLR with cross-entropy loss and a positive coefficient $\lambda$ for $\ell_2$ regularization creates a strongly convex objective that satisfies the Assumption 3.1 (proofs are in Appendix A.4). This allows the exact calculation of the gradients w.r.t. the sample weights via Equation 9. Secondly, calculating the Hessian inverse in Equation 6 is no longer overly expensive, because the only free model parameters are in the last layer instead of the entire deep neural network. Thirdly, LLR acts as a regularizer by limiting the capacity of the model parameters, which counterintuitively facilitates the training with weighted objectives. This is because weighted objectives do not necessarily yield different predictors compared to ERM when training overparameterized models as shown in Zhai et al. (2022). Sufficient regularization is usually needed for the reweighting scheme to be practically meaningful (Byrd & Lipton, 2019; Sagawa et al., 2019). Therefore, regularized LLR ensures that the theoretical soundness of our method translates into practical benefits without violating assumptions or conducting extensive approximations.

## 4.2 Our GSR Algorithm

A comprehensive overview of GSR is presented in Algorithm 1. For simplicity, we omit certain details such as regularization and gradient clipping. Specifically, it contains two stages:

**(Stage 1) Representation learning.** We take out a random subset (e.g., 10%) from the training set as a held-out set $\mathcal{D}^{\text{tr-h}}$. Then, train the entire model $\theta$ on the remaining training set $\mathcal{D}^{\text{tr-r}}$ with ERM until convergence without early stopping. This ensures that the model fits the "remaining" training set $\mathcal{D}^{\text{tr-r}}$ well. Reserving a held-out set $\mathcal{D}^{\text{tr-h}}$ that the model has not encountered during training is a crucial step, as it prevents overfitting to the data that will be later used for sample reweighting. Otherwise, changing the sample weights will not make a meaningful difference to the last-layer classifier (Zhai et al., 2022). Note that neither group labels nor sample weights are required at this stage.

**(Stage 2) Group-robust sample reweighting with last-layer retraining.** Denote the model parameters as $\theta = (\phi, \psi)$ where $\phi$ is the feature extractor and $\psi$ is the last-layer linear classifier. In this stage, we keep the feature extractor $\phi$ fixed, and jointly train the last layer $\psi$ and the sample weights $w$ using the held-out set $\mathcal{D}^{\text{tr-h}}$ (as a training set) and the target set $\mathcal{D}^{\text{tar}}$. Subsequently, we perform model selection with the validation set $\mathcal{D}^{\text{v}}$. We create $\mathcal{D}^{\text{tar}}$ and $\mathcal{D}^{\text{v}}$ with equal size, by splitting the

---

**Algorithm 1** Group-robust Sample Reweighting with last-layer retraining (`GSR`)

---

**Require:** Training set $\mathcal{D}^{\text{tr}}$ of size $n$, validation set $\mathcal{D}^{\text{v}}$, target set $\mathcal{D}^{\text{tar}}$ with $m$ groups, held-out set fraction $\alpha$, outer loop steps $T$, outer learning rate $\beta$, scaling temperature $\tau$.
1: Obtain the held-out set $\mathcal{D}^{\text{tr-h}}$ with size $\alpha n$ and the remaining set $\mathcal{D}^{\text{tr-r}}$ with size $(1-\alpha)n$
    **Stage 1:**
2: $(\phi^*, \psi^*) \leftarrow \arg\min_{\theta=(\phi,\psi)} \hat{\mathcal{R}}(\mathcal{D}^{\text{tr-r}}; \theta)$
    **Stage 2:**
3: Initialize held-out sample weights $w^{(1)} \leftarrow \frac{1}{\alpha n} \cdot \mathbf{1}^{\alpha n}$
4: Initialize adaptive influence score weights $\gamma^{(0)} \leftarrow \frac{1}{m} \cdot \mathbf{1}^m$
5: **for** $t = 1, \ldots, T$ **do**
6:    $\psi^{(t)} \leftarrow \arg\min_{\psi} \hat{\mathcal{R}}(\mathcal{D}^{\text{tr-h}}; w^{(t)}, (\phi^*, \psi))$               $\triangleright$ Last-layer retraining
7:    **for** $g = 1, \ldots, m$ **do**
8:       $\gamma_g^{(t)} \leftarrow \gamma_g^{(t-1)} \exp\left(\hat{\mathcal{R}}(\mathcal{D}_g^{\text{tar}}; (\phi^*, \psi^{(t)}))/\tau\right)$      $\triangleright$ Update aggregation weights
9:    **end for**
10:   $\gamma^{(t)} \leftarrow \gamma^{(t)}/\|\gamma^{(t)}\|_1$
11:   $\xi^{(t)} \leftarrow \mathbf{0}^{\alpha n}$
12:   **for** $g = 1, \ldots, m$ **do**
13:      $\xi^{(t)} = \xi^{(t)} + \gamma_g^{(t)}\widetilde{\mathcal{I}}(\mathcal{D}^{\text{tr-h}}, \mathcal{D}_g^{\text{tar}}; (\phi^*, \psi^{(t)}), w)$    $\triangleright$ Adaptive aggregation of the influence
14:   **end for**
15:   $w^{(t+1)} \leftarrow \max\{w^{(t)} - \beta\xi^{(t)}, \mathbf{0}^{\alpha n}\}$        $\triangleright$ Projected gradient descent (element-wise max)
16:   $w^{(t+1)} \leftarrow w^{(t+1)}/\|w^{(t+1)}\|_1$            $\triangleright$ Weight normalization
17:   **if** $\hat{\mathcal{R}}^{\text{WG}}(\mathcal{D}^{\text{v}}; (\phi^*, \psi^{(t)})) \leq \hat{\mathcal{R}}^{\text{WG}}(\mathcal{D}^{\text{v}}; (\phi^*, \psi^*))$ **then**
18:      $\psi^* \leftarrow \psi^{(t)}$                     $\triangleright$ Model selection
19:   **end if**
20: **end for**
21: **return** $\theta^* \leftarrow (\phi^*, \psi^*)$

---

initial "standard" validation set (as $\mathcal{D}^{\text{L}}$) in half (see remark below for the reason). The inner loop, which fits a linear classifier $\psi$ on the weighted cross-entropy objective with $\ell_2$ regularization, is optimized via L-BFGS (Liu & Nocedal, 1989) for efficiency. The outer loop performs projected gradient descent with learning rate $\beta$. The gradient is calculated using adaptive aggregation of the influence scores (Equation 9), projected to $\{w|w_i \geq 0\,\forall i\}$. The Hessian of the inner objective is updated as $H_{\psi^{(t)}, w^{(t)}} = \nabla_\psi^2 \hat{\mathcal{R}}(\mathcal{D}^{\text{tr-h}}; w^{(t)}, \psi^{(t)})$ in each iteration. In addition, the gradient of the sample weights can become spiky when high loss values are encountered on the validation set. We employ gradient clipping by norm and weights normalization for better training stability.

**Remark.** The target set and the validation set cannot be the same. Although the target data is only used to update the weights of the held-out training data and does not directly affect the training of the model parameters, overfitting to the target can still occur as a result of the representer theorem (Schölkopf et al., 2001). Thus, it is essential to create different splits for the target and validation set to prevent overfitting during sample reweighting, which we illustrate later in Figure 4d.

## 5 EXPERIMENTS

**Datasets.** We evaluate the effectiveness of algorithms on 4 commonly used datasets. *Waterbirds* (Wah et al., 2011) is a binary object recognition dataset for bird types (i.e., waterbird, landbird), which are spuriously correlated with the background (i.e., water, land). *CelebA* (Liu et al., 2015) is a binary object recognition dataset for hair color blondness prediction. There exists a spurious correlation between the man gender attribute and non-blondness. *MultiNLI* (Williams et al., 2017) is a multi-class natural language inference dataset. The three classes (i.e., entailment, neutral, contradiction) describe the relationship between a pair of sentences. The spurious correlation exists between contradiction and the presence of negation words. *CivilComments (WILDS)* (Borkan et al., 2019; Koh et al., 2021) is a binary text toxicity detection dataset. There are 8 types of identities mentioned in the text, such as male and female. Grouping the samples by the identity and the class results in 16 overlapping groups. Although there are no obvious spurious correlations, the data is extremely imbalanced among

Table 1: **Performance comparison.** We report the worst-group accuracy on four benchmark datasets. For the group label column, ✗ indicates no group label $g$ is used; ✓ indicates $g$ is required; ✓̷ indicates a proportion of $g$ from the validation set is re-purposed beyond hyperparameter tuning, such as training sample weights or model parameters. The row sections are ranked in descending order of the total amount of group label information required. We report the mean $\pm$ standard deviation over 5 random seeds. We use "$-$" to indicate missing evaluation results from the original paper.

| Method | Group label Train / Val | Worst-group accuracy (%) | | | | Average |
|---|---|---|---|---|---|---|
| | | Waterbirds | CelebA | MultiNLI | CivilComments | |
| Group DRO | ✓/ ✓ | $91.4_{\pm1.1}$ | $88.9_{\pm2.3}$ | $77.7_{\pm1.4}$ | $70.0_{\pm2.0}$ | 82.0 |
| RWG | ✓/ ✓ | $87.6_{\pm1.6}$ | $84.3_{\pm1.8}$ | $69.6_{\pm1.0}$ | $72.0_{\pm1.9}$ | 78.4 |
| JTT | ✗/ ✓ | 86.7 | 81.1 | 72.6 | 69.3 | 77.4 |
| CnC | ✗/ ✓ | $88.5_{\pm0.3}$ | $88.8_{\pm0.9}$ | $-$ | $68.9_{\pm2.1}$ | $-$ |
| SSA | ✗/ ✓̷ | $89.0_{\pm0.6}$ | $\mathbf{89.8}_{\pm1.3}$ | $76.6_{\pm0.7}$ | $69.9_{\pm2.0}$ | 81.3 |
| MAPLE | ✗/ ✓̷ | 91.7 | 88.0 | 72.7 | 64.1 | 79.1 |
| DFR | ✗/ ✓̷ | $\mathbf{92.9}_{\pm0.2}$ | $88.3_{\pm1.1}$ | $74.7_{\pm0.7}$ | $70.1_{\pm0.8}$ | 81.5 |
| GSR−HF | ✗/ ✓̷ | $87.5_{\pm0.1}$ | $86.3_{\pm0.4}$ | $74.7_{\pm0.0}$ | $68.9_{\pm0.2}$ | 79.4 |
| GSR | ✗/ ✓̷ | $\mathbf{92.9}_{\pm0.0}$ | $87.0_{\pm0.4}$ | $\mathbf{78.5}_{\pm0.3}$ | $\mathbf{71.7}_{\pm0.6}$ | $\mathbf{82.5}$ |
| ERM | ✗/ ✗ | $74.9_{\pm2.4}$ | $46.9_{\pm2.8}$ | $65.9_{\pm0.3}$ | $55.6_{\pm0.6}$ | 60.8 |
| SELF | ✗/ ✗ | $\mathbf{93.0}_{\pm0.3}$ | $83.9_{\pm0.9}$ | $70.7_{\pm2.5}$ | $-$ | $-$ |

these groups, especially on text that mentions other religions and is classified as non-toxic. Overall, we follow the standard train, validation, and test splits for all datasets. We randomly create target and validation sets by equally splitting the original validation.

**Baselines.** We compare our approach against baselines in three categories according to the amount of group labels required. (1) Group labels for both training and validation: Group DRO (Sagawa et al., 2019) optimizes the worst-group training loss by dynamically adjusting the group weights; RWG (Idrissi et al., 2022) balances the sampling probability of each group according to their sizes. (2) Group labels for validation: JTT (Liu et al., 2021) upweights the high-loss training points that are more likely to be from minority groups. CnC (Zhang et al., 2022) in addition uses contrastive learning to align the representations of the same-class samples. SSA (Nam et al., 2022) infers the pseudo group labels of the training set by training a predictor on the validation set. DFR (Kirichenko et al., 2022) performs group-balanced last-layer retraining on the validation set. MAPLE[1] (Zhou et al., 2022) is the most similar to ours that uses the validation set to jointly reweight the training samples and retrain the entire model. (3) No group labels: SELF (LaBonte et al., 2024) constructs an approximately group-balanced dataset based on the disagreement from an auxiliary model, then performs class-balanced last-layer retraining. We choose the best-performing ES disagreement SELF.

**Setup.** For GSR, at stage 1, we use the suggested hyperparameter configuration and data augmentation strategies from DFR on all datasets (with minor modifications for MultiNLI and CivilComments). At stage 2, we drop data augmentation and perform a randomized search of hyperparameters using the valuation set $\mathcal{D}^v$. The details are documented in Appendix B.2.

## 5.1 IMPROVEMENT IN GROUP ROBUSTNESS

The main results are in Table 1. Our approach GSR achieves consistent and competitive results compared to the baseline methods in terms of group robustness, which is measured by the worst-group accuracy. Specifically, GSR has the state-of-the-art (SoTA) performance for both the MultiNLI and CivilComments datasets, and close-to-SoTA worst-group accuracy on the Waterbirds and CelebA datasets. Moreover, GSR demonstrated the highest consistency across all four datasets. It achieves an average improvement of 1.0% in absolute worst-group accuracy compared to DFR (the SoTA method which uses the same amount of group labels as ours), because GSR allows more fine-grained reweighting over the training data. GSR even outperforms Group DRO which requires more group labels, indicating that training the full network with weighted data may not always be necessary. The detailed results including the average accuracy of the methods are in Appendix B.5.

---

[1]We re-implemented MAPLE with better efficiency and support for MultiNLI and CivilComments.

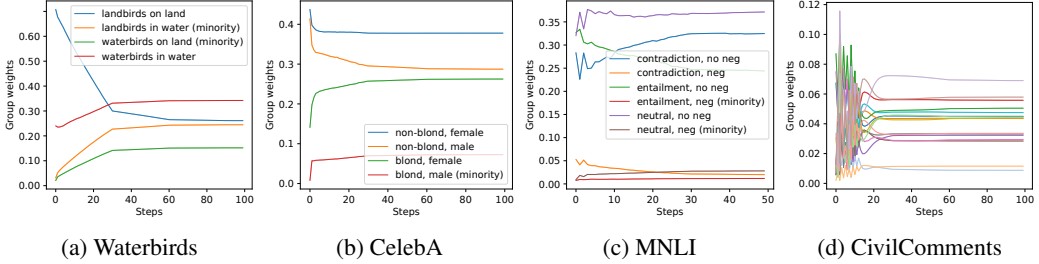

Figure 1: The change in the sum of sample weights across different groups throughout the training. The minority groups are upweighted and the majority groups are generally downweighted. However, different groups do not have equal sums of weights.

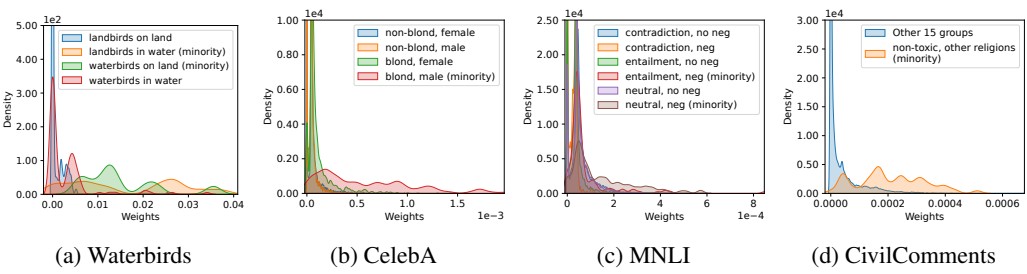

Figure 2: The distribution of sample weights for each group that are used to train the best models. The minority groups have the weight distribution stretched out towards high values, while the majority-group weights are generally skewed towards 0.

Besides, as an ablation study, we include the Hessian-free version of our method, `GSR-HF`, which uses a simplified sample weight update from MAPLE (Equation 3) during the last-layer retraining. As shown in Table 1, `GSR-HF` performs consistently worse than `GSR`, validating our analysis in Section 2. This highlights the importance of incorporating the Hessian matrix in Equation 6 to ensure the correctness of the gradient when optimizing the sample weights.

## 5.2 A CLOSER LOOK AT SAMPLE WEIGHTS

**Varying weights across groups.** Figure 1 illustrates the variation in the sum of sample weights across different groups through the training process of the best-performing model reported in Table 1. In general, all the minority-group weights increase, and the majority groups with spurious correlations generally have decreased weights. For example, the weights for minority groups (e.g., landbirds in water in Figure 1a) increase as optimization progresses. In contrast, the weights for certain majority groups with clear spurious correlations (e.g., landbirds on land in Figure 1a) decrease over the optimization steps. Note that not all majority groups necessarily experience a decrease in their sum of sample weights. Another key message here is that the better-performing models are not necessarily trained with equally weighted groups. In all cases, the worst-group performance is better optimized even though the groups are weighted differently, which effectively mitigates the spurious correlations with still "imbalanced" data. These results obtained from directly optimizing the sample weights align well with the similar findings in Qiu et al. (2023); Sagawa et al. (2019).

**Varying weights within groups.** We visualize the distribution of the sample weights that are used to train the best models in Figure 2 (where the distributions are smoothed using kernel density estimation). The samples from the majority groups are consistently assigned with lower weights after reweighting, especially for Waterbirds and CelebA in Figures 2a, 2b. On the other hand, the minority groups are consistently upweighted with their sample weights stretched out to larger values across all datasets. We can also clearly see that our best-performing models are trained on samples with diverse weights, even if they are from the same group. This implies that not all samples need to be upweighted or downweighted for group-balanced training. Having more fine-grained adjustments over the sample weights offers more flexibility in improving the robustness to subpopulation shifts.

**Visualizing high-weight samples.** In addition, we visualize the held-out samples with the highest weight from Waterbirds in Figure 3. These high-weight samples also tend to be consistent across different runs. All of these images are from the minority groups (waterbirds on land, landbirds in water). As highlighted by the red box, the image has an ambiguous background that blends features of both land and water. Hence, it should be considered as a minority. Since the group labels for the held-out sets are not used for retraining, GSR is not affected by the annotation errors for group labels and correctly assigns high weights to these samples, since they are supposedly helpful in improving the worst-group performance on the target set. However, other group-balanced baselines that directly train on these group labels do not offer such a benefit. We will subsequently show that GSR is robust to class-label errors in the training data in Section 5.3.

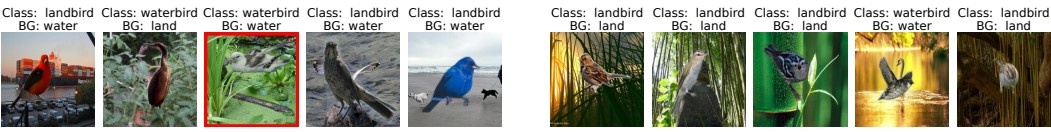

(a) The *most* weighted instances.      (b) The *least* weighted instances.

Figure 3: We illustrate the selected images with their class label and background label from the held-out split in Waterbirds according to the sample weights. In 3a, the top-5 *most* weighted instances are all from the minority groups with differing bird types and backgrounds. As highlighted in the red box in (a), the background of the waterbird is ambiguous as it blends features of both land and water and hence should be categorized as a minority. GSR correctly identifies it despite the suboptimal annotation. In 3b, the top-5 *least* weighted instances are all from the majority groups where the background is spuriously correlated with the bird type.

## 5.3 ROBUSTNESS TO LABEL NOISE

The real-world data often have annotation errors (Beyer et al., 2020; Gururangan et al., 2018). If not addressed properly, the errors can substantially degrade the model's group robustness when the spurious correlation is strong in the training set (Qiao & Low, 2024). Assuming efforts are invested into the quality of the group-labeled dataset $\mathcal{D}^{\mathrm{L}}$, the other training data without group labels in $\mathcal{D}^{\mathrm{U}}$ may have unaddressed errors such as mislabeled classes. To evaluate the resilience of GSR on training data with different quality, we simulate the variation of data quality by adding class label noise. In particular, we randomly flip up to 40% of the class labels in the held-out set to incorrect classes, and leave the target and validation sets unchanged. As shown in Figure 4a, the worst-group performance of GSR has minimal performance degradation. This is because using high-quality targets for training sample reweighting allows more fine-grained control over the training data. To further understand why, we plot the learned weight distribution of the held-out data in Waterbirds. Compared to that of the clean data in Figure 4b, the weight distribution of the noisy data is more skewed towards 0 as shown in Figure 4c. In particular, almost all minority instances with noisy labels have close-to-0

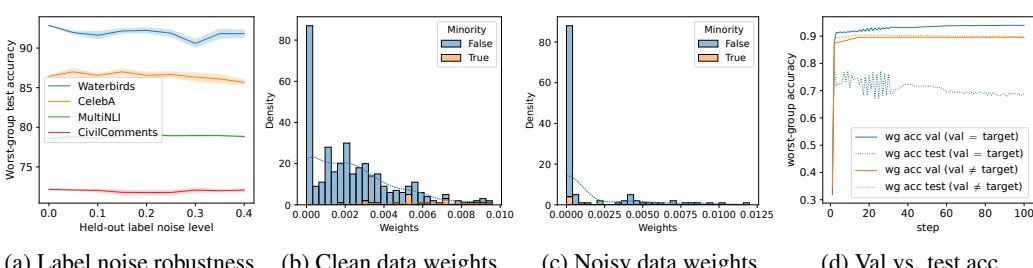

(a) Label noise robustness      (b) Clean data weights      (c) Noisy data weights      (d) Val vs. test acc

Figure 4: In-depth study of our algorithm. In 4a, the worst-group test accuracy degrades slightly even when up to 40% of the held-out set labels are corrupted. In 4b, most of the uncorrupted minority samples received higher weight assignments than non-minority examples. In contrast, in 4c, the corrupted minority instances are correctly assigned with close-to-0 weights. 4d shows the relationship between validation and test worst-group accuracy on the CelebA dataset. It is important to have separate target and validation sets. Otherwise, overfitting can easily occur.

weights, equivalent to being removed from the training set. Therefore, the efficacy of GSR can be attributed to the property of automatically cleaning the training data via sample reweighting, achieving a similar effect as other noisy label detection and removal approaches (Ghorbani & Zou, 2019; Wang et al., 2024b). GSR being robust to class label corruption has an important implication: to improve the robustness to distribution shifts under a limited annotation budget, more efforts should be dedicated to ensuring the quality of the target set.

## 6  RELATED WORK

**Subpopulation shifts.** Group labels are essential for training and evaluating machine learning models under subpopulation shifts. Various approaches have been proposed to handle scenarios with different levels of availability of group labels: fully known for all data splits (Deng et al., 2024; Idrissi et al., 2022; Sagawa et al., 2019), partially known (Kirichenko et al., 2022; Liu et al., 2021; 2022; Nam et al., 2022; Qiu et al., 2023; Zhang et al., 2022; Zhou et al., 2022), or completely unknown (Han & Zou, 2024; LaBonte et al., 2024). In addition, Yang et al. (2023) comprehensively studied the effectiveness of these approaches on a variety of benchmarks with unknown group labels and demonstrated that optimizing for the worst class can be surprisingly effective for improving the worst-group performance. Our approach falls into the category of learning with partial group labels (from the validation set) and differs from most of the approaches by using group-labeled data for sample reweighting instead of direct training.

**Sample reweighting and implicit differentiation.** Importance weighting is a classic paradigm for learning under distribution shifts (Chen et al., 2025; Fang et al., 2020; 2023). The idea can be further expanded to bilevel optimization of the sample weights and the model parameters (Ren et al., 2018; Zhou et al., 2022). These works on bilevel optimization focus on using truncated backpropagation to circumvent the challenge of differentiating through the unrolled training trajectory. Implicit differentiation is an alternative solution to the problem and has been applied to areas such as hyperparameter optimization (Foo et al., 2007; Lorraine et al., 2020; Pedregosa, 2016), meta-learning (Lee et al., 2019; Rajeswaran et al., 2019), generative modeling (Domke, 2012; Samuel & Tappen, 2009), and fairness (Shui et al., 2022). In addition, Li & Liu (2022); Wang et al. (2024a) have explored one-step (i.e., not iterative) sample reweighting via the influence function for algorithmic fairness. Bae et al. (2022) shows an alternative derivation of the influence function via implicit differentiation on the training set. In contrast to these works, we show the connection between the influence function and bilevel optimization via implicit differentiation and devise an efficient scheme to rigorously apply it iteratively for addressing subpopulation shifts.

## 7  CONCLUSION AND FUTURE WORK

Motivated from the bilevel minimax objective, GSR utilizes the group-labeled data for fine-grained reweighting of the training samples via a soft aggregation of the influence functions. It seamlessly integrates with last-layer retraining, resulting in a lightweight and effective strategy for improving group robustness. GSR also has the ability to automatically clean and guide the training of potentially noisy datasets via sample reweighting. Our empirical analysis further supports the alternative training paradigm that emphasizes the use of high-quality, group-labeled instances as the target set.

We identify several future research directions to further improve our method. Despite GSR achieving competitive worst-group performance, it results in a decrease in mean performance. This trade-off is common in efforts to improve group robustness (Chen et al., 2022). Additionally, our work is based on the hypothesis that standard ERM learns sufficiently high-quality representations. Although GSR test the limit of last-layer retraining, it does not directly improve the representation learning. To mitigate the trade-off between the worst-group performance and the mean performance, enhancing the quality of representations is still a critical step (Izmailov et al., 2022). To answer the question of whether sample reweighting can improve the representation learning for the group robustness, it is necessary to retrain more components of the model after obtaining the sample weights. Developing more efficient strategies that jointly optimize the sample weights and deep model parameters may offer more insights and advancements in tackling subpopulation shifts and beyond.

ACKNOWLEDGEMENT

This research/project is supported by the National Research Foundation, Singapore under its AI Singapore Programme (AISG Award No: AISG2-PhD/2021-08-017[T]). This research is supported by the National Research Foundation Singapore and the Singapore Ministry of Digital Development and Innovation, National AI Group under the AI Visiting Professorship Programme (award number AIVP-2024-001). This research is supported by the National Research Foundation (NRF), Prime Minister's Office, Singapore under its Campus for Research Excellence and Technological Enterprise (CREATE) programme. The Mens, Manus, and Machina (M3S) is an interdisciplinary research group (IRG) of the Singapore MIT Alliance for Research and Technology (SMART) centre. Jingtan Wang is supported by the Institute for Infocomm Research of Agency for Science, Technology and Research (A*STAR). We would like to thank the anonymous reviewers and AC for the constructive and helpful feedback.

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

## A  DERIVATIONS AND PROOFS

### A.1  DERIVATION OF THE JACOBIAN OF THE GRADIENT W.R.T. THE SAMPLE WEIGHTS

The derivative of $\nabla_\theta \hat{\mathcal{R}}(\mathcal{D}^{\text{tr}}; w, \theta)$ w.r.t. individual sample weights $w_i$ is:

$$
\begin{aligned}
\nabla_{w_i} \nabla_\theta \hat{\mathcal{R}}(\mathcal{D}^{\text{tr}}; w, \theta) &= \nabla_{w_i} \nabla_\theta \sum_{j \in 1, \dots, |\mathcal{D}^{\text{tr}}|} w_j \ell(x_j^{\text{tr}}, y_j^{\text{tr}}; \theta) \\
&= \nabla_{w_i} \sum_{j \in 1, \dots, |\mathcal{D}^{\text{tr}}|} w_j \nabla_\theta \ell(x_j^{\text{tr}}, y_j^{\text{tr}}; \theta) \\
&= \nabla_\theta \ell(x_i^{\text{tr}}, y_i^{\text{tr}}; \theta) \\
&= \nabla_\theta \hat{\mathcal{R}}(z_i; \theta_T) .
\end{aligned}
\tag{10}
$$

### A.2  DERIVATION OF LAST-STEP GRADIENT (MAPLE (ZHOU ET AL., 2022))

According to MAPLE's one-step truncated backpropagation, the gradient descent updates before $\theta_{T-1}$ are truncated, so that $\theta_{T-1}$ is a "constant" and is independent of $w$. Thus, it is true that $\nabla_{w_i} \theta_{T-1} = 0$, but not for $\nabla_{w_i} \theta_T$. Recall that for simplicity, we use the notation $\nabla_\theta \hat{\mathcal{R}}(\mathcal{D}^{\text{tr}}; w, \theta_T)$ for $\nabla_\theta \hat{\mathcal{R}}(\mathcal{D}^{\text{tr}}; w, \theta)\big|_{\theta=\theta_T}$. We show that Equation 2 can be derived as follows:

$$
\begin{aligned}
\nabla_{w_i} \hat{\theta}^* &\approx \nabla_{w_i} \theta_T \\
&\approx \nabla_{w_i} \left( \theta_{T-1} - \eta \nabla_\theta \hat{\mathcal{R}}(\mathcal{D}^{\text{tr}}; w, \theta_{T-1}) \right) \qquad \text{(One-step truncation)} \\
&\approx \nabla_{w_i} \left( \theta_{T-1} - \eta \nabla_\theta \hat{\mathcal{R}}(\mathcal{D}^{\text{tr}}; w, \theta_T) \right) \\
&\qquad (\theta_T \approx \theta_{T-1} \text{ on convergence, so } \nabla_\theta \hat{\mathcal{R}}(\mathcal{D}^{\text{tr}}; w, \theta_T) \approx \nabla_\theta \hat{\mathcal{R}}(\mathcal{D}^{\text{tr}}; w, \theta_{T-1})) \\
&= 0 - \eta \nabla_{w_i} \nabla_\theta \hat{\mathcal{R}}(\mathcal{D}^{\text{tr}}; w, \theta_T) \qquad (\nabla_{w_i} \theta_{T-1} = 0 \text{ because of one-step truncation}) \\
&= -\eta \nabla_\theta \hat{\mathcal{R}}(z_i; \theta_T) . \qquad \text{(Apply Equation 10)}
\end{aligned}
$$

### A.3  DERIVATION OF META-GRADIENT UPDATE

We utilize the technique of implicit differentiation. Upon convergence to an optimal solution, the inner problem has the property of $\nabla_\theta \hat{\mathcal{R}}(\mathcal{D}^{\text{tr}}; w, \hat{\theta}^*) = 0$. By first differentiating w.r.t. $w$ on both sides of the equation, then followed by the chain rule, we have the following derivations:

$$
\begin{aligned}
\frac{\mathrm{d}0}{\mathrm{d}w} &= \frac{\mathrm{d}\nabla_\theta \hat{\mathcal{R}}(\mathcal{D}^{\text{tr}}; w, \hat{\theta}^*)}{\mathrm{d}w} \\
0 &= \frac{\mathrm{d}\nabla_\theta \hat{\mathcal{R}}(\mathcal{D}^{\text{tr}}; w, \hat{\theta}^*)}{\mathrm{d}w} \\
0 &= \frac{\partial \nabla_\theta \hat{\mathcal{R}}(\mathcal{D}^{\text{tr}}; w, \hat{\theta}^*)}{\partial w} + \frac{\partial \nabla_\theta \hat{\mathcal{R}}(\mathcal{D}^{\text{tr}}; w, \hat{\theta}^*)}{\partial \hat{\theta}^*} \frac{\mathrm{d}\hat{\theta}^*}{\mathrm{d}w} \qquad \text{(Chain-rule)} \\
0 &= \nabla_w \nabla_\theta \hat{\mathcal{R}}(\mathcal{D}^{\text{tr}}; w, \hat{\theta}^*) + \nabla_\theta^2 \hat{\mathcal{R}}(\mathcal{D}^{\text{tr}}; w, \hat{\theta}^*) \frac{\mathrm{d}\hat{\theta}^*}{\mathrm{d}w} \\
\frac{\mathrm{d}\hat{\theta}^*}{\mathrm{d}w} &= - \left( \nabla_\theta^2 \hat{\mathcal{R}}(\mathcal{D}^{\text{tr}}; w, \hat{\theta}^*) \right)^{-1} \nabla_w \nabla_\theta \hat{\mathcal{R}}(\mathcal{D}^{\text{tr}}; w, \hat{\theta}^*) \\
\frac{\mathrm{d}\hat{\theta}^*}{\mathrm{d}w} &= -H_{\hat{\theta}^*, w}^{-1} \nabla_w \nabla_\theta \hat{\mathcal{R}}(\mathcal{D}^{\text{tr}}; w, \hat{\theta}^*) .
\end{aligned}
$$

For each sample weight $w_i$:

$$
\begin{aligned}
\frac{\mathrm{d}\hat{\theta}^*}{\mathrm{d}w_i} &= -H_{\hat{\theta}^*, w}^{-1} \nabla_{w_i} \nabla_\theta \hat{\mathcal{R}}(\mathcal{D}^{\text{tr}}; w, \hat{\theta}^*) \\
&= -H_{\hat{\theta}^*, w}^{-1} \hat{\mathcal{R}}(z_i; \theta_T) . \qquad \text{(Apply Equation 10)}
\end{aligned}
$$

### A.4 PROOF OF THE STRONG CONVEXITY OF LINEAR CLASSIFICATION WITH CROSS-ENTROPY LOSS AND $\ell_2$-REGULARIZATION

In this section, we add the complete proof for the established result of the strong convexity (a subset of strict convexity) of linear classification with cross-entropy loss and $\ell_2$-regularization. Before introducing the complete proof, a shortcut intuitive proof for this consists of the following steps:

1. Linear classification with cross-entropy loss is convex, and its Hessian $H$ is positive semi-definite, implying $H \succeq 0$.
2. Adding $\ell_2$-regularization with a positive regularization strength $\lambda > 0$ produces a new Hessian $H' = H + \lambda I \succeq \lambda I$, which meets the definition of strong convexity (Definition A.1).

We now present a complete proof of the result of the strong convexity by directly showing the $H'$ in Theorem A.6.

**Definition A.1** ($\mu$-Strong convexity (Boyd & Vandenberghe, 2004)). For $\mu > 0$, a differentiable function $f$ is $\mu$-strongly convex if for some $\mu > 0$, $\nabla^2 f(x) \succeq \mu I$.

**Definition A.2** (One-hot Vector). For a dimension $d$, $u(i) \in \{0,1\}^d, \sum u(i) = 1$ is the one-hot vector with $i$-th dimension $u(i)_i = 1$. We omit $d$ in the following context for simplicity.

**Definition A.3** (Softmax). Given the input $x \in \mathbb{R}^d$, define the softmax function $\sigma : \mathbb{R}^d \to \mathbb{R}^d$ as:

$$\sigma(x)_i = \frac{e^{x_i}}{\sum_i e^{x_i}} .$$

Its derivative $\nabla_x \sigma(x) \in \mathbb{R}^{d \times d}$ has the form:

$$\nabla_x \sigma(x) = \text{diag}(\sigma(x)) - \sigma(x)\sigma(x)^\top . \tag{11}$$

In addition,

$$\nabla_x \sigma(x)_i = \sigma(x)_i (u(i) - \sigma(x)) , \tag{12}$$

where $u(i) \in \{0,1\}^d, \sum u(i) = 1$ is the one-hot vector with $i$-th dimension $u(i)_i = 1$.

**Proposition A.4.** *The first-order derivative matrix $\nabla_x \sigma(x)$ of the softmax function is positive semidefinite.*

*Proof.* First $\sigma(x)$ defines a probability distribution. Hence, it has the property that $\sigma(x)_i \geq 0 \ \forall i$, $\sum_i \sigma(x)_i = 1$. Then according to the definition of positive semidefiniteness: $\forall z \in \mathbb{R}^d, z \neq \mathbf{0}$:

$$\begin{aligned}
z^\top \nabla_x \sigma(x) z &= z^\top (\text{diag}(\sigma(x)) - \sigma(x)\sigma(x)^\top) z \\
&= z^\top \text{diag}(\sigma(x)) z - z^\top \sigma(x)\sigma(x)^\top z \\
&= \sum_i \sigma(x)_i z_i^2 - (\sum_i \sigma(x)_i z_i)^2 \\
&\geq 0 . \qquad\qquad \text{(Cauchy-Schwartz)}
\end{aligned}$$

$\square$

**Definition A.5** (Cross-entropy). Given the input $z \in \mathbb{R}^d, y \in \{1, \dots, K\}$ for a $K$-way classification problem, define the cross entropy loss $L$:

$$\text{CE}(z, y) = - \sum_k^K \mathbb{I}[y = k] \log z_i ,$$

where $\mathbb{I}$ is the indicator function.

**Theorem A.6** (Strong convexity of cross-entropy loss with $\ell_2$-regularization). *For multi-class linear classification with cross-entropy loss, if $\ell_2$-regularization with positive coefficient $\lambda$ is applied, then the objective is $\lambda$-strongly convex.*

*Proof.* Let $x \in \mathbb{R}, y \in \{1, \ldots, K\}$ denote the input, label of a $K$-way classification task. Let $\psi \in \mathbb{R}^{d \times K}$ denote the linear model parameters. Let $u(y)$ denote the one-hot representation of $y$. The objective function using cross-entropy loss and $\ell_2$-regularization is:

$$L(x, y; \psi) = \text{CE}(\sigma(\psi^\top x), y) + \frac{\lambda}{2} \|\psi\|_2^2$$

$$= -\sum_k^K \mathbb{I}[y = k] \log\big(\sigma(\psi^\top x)_k\big) + \frac{\lambda}{2} \|\psi\|_2^2$$

$$= -\log\big(\sigma(\psi^\top x)_y\big) + \frac{\lambda}{2} \|\psi\|_2^2 \,.$$

The first-order derivative:

$$\nabla_\psi L(x, y; \psi) = -x \frac{1}{\sigma(\psi^\top x)_y} \sigma(\psi^\top x)_y \big(u(y) - \sigma(\psi^\top x)\big)^\top + \lambda\psi \qquad \text{(Equation 12)}$$

$$= x(\sigma(\psi^\top x) - u(y))^\top + \lambda\psi \,.$$

The second-order derivative (Hessian):

$$H = \nabla_\psi^2 L(x, y; \psi)$$

$$= \frac{\partial x(\sigma(\psi^\top x) - u(y))^\top}{\partial \psi} + \lambda I_{d \times k}$$

$$= x \frac{\partial \sigma(\psi^\top x)^\top}{\partial \psi} + \lambda I_{d \times k}$$

$$= x \frac{\partial \sigma(\psi^\top x)^\top}{\partial (\psi^\top x)} \frac{\partial \psi^\top x}{\partial \psi} + \lambda I_{d \times k}$$

$$= x[x(\text{diag}(\sigma(\psi^\top x)) - \sigma(\psi^\top x)\sigma(\psi^\top x)^\top)]^\top + \lambda I_{d \times k} \qquad \text{(Equation 11)}$$

$$= x \left(\text{diag}(\sigma(\psi^\top x)) - \sigma(\psi^\top x)\sigma(\psi^\top x)^\top\right) x^\top + \lambda I_{d \times k} \,.$$

Then we show that the Hessian of $L$ meets the requirement of $\lambda$-strong convexity, i.e., $H - \lambda I \succeq 0$, $\forall z \in \mathbb{R}^{d \times k}, z \neq 0$:

$$z^\top (H - \lambda I_{d \times k}) z = z^\top x \left(\text{diag}(\sigma(\psi^\top x)) - \sigma(\psi^\top x)\sigma(\psi^\top x)^\top\right) x^\top z + z^\top (\lambda I_{d \times k} - \lambda I_{d \times k}) z$$

$$= z^\top x \left(\text{diag}(\sigma(\psi^\top x)) - \sigma(\psi^\top x)\sigma(\psi^\top x)^\top\right) x^\top z$$

$$\geq 0 \,. \qquad \text{(Proposition A.4)}$$

Hence, $L$ is strongly convex according to Definition A.1. $\qquad \square$

# B EXPERIMENTAL DETAILS

## B.1 SETUP

For model architectures and initialization, we use ImageNet-pretrained ResNet-50 (V1) (He et al., 2016) for Waterbirds and CelebA datasets, and use BERT (HuggingFace) for MultiNLI and Civil-Comments. The MNLI dataset is preprocessed according to the setup in `https://github.com/kohpangwei/group_DROBertTokenizer`. The CivilComment dataset is tokenized with Bert-Tokenizer with max_seq_length=300. For all experiments, we perform 5 independent runs on seed [1, 2, 3, 4, 5] to obtain the mean and the standard deviations.

## B.2 HYPERPARAMETERS

For all experiments, we fix the held-out fraction $\alpha$ as 10% and fix the resulting partition splits (i.e., the held-out set and the remaining set) of the training set $\mathcal{D}^{\text{tr}}$, which are generated by setting the

numpy random seed to 1. Although additionally tuning the held-out fraction $\alpha$ and changing the split can most likely improve the worst-group performance by balancing the data for representation learning and sample reweighting, we do not alter these splits for better consistency and only focus on the second sample reweighting stage of the training.

For stage 1, we use *almost* the same hyperparameters as used by DFR (some are slightly altered for better consistency and simplicity by heuristics *without* any tuning) and record them in Table 2.

For stage 2, there are two sets of hyperparameters for the inner and the outer loops. We perform a non-exhaustive search by randomly sampling the combination of hyperparameters from the grid. We select the best-performing model checkpoints and hyperparameter configurations based on the worst-group validation accuracy. The selected hyperparameter configurations are documented in Tables 3 and 4 for the outer loop and the inner loop respectively. Besides, we consistently apply weight normalization (to ensure the sum of weights equals one) and a step learning rate scheduler (StepLR), which decays the outer learning rate by a factor of 10 every 30 steps. This facilitates a better convergence for the bilevel optimization.

Table 2: **Hyperparameters for base model training**. For Waterbirds and CelebA, we use *almost* the same hyperparameters as used by DFR. We slightly change the hyperparameters on MNLI and CivilComments (without any tuning) so that no learning rate scheduler is used. Momentum is set as 0.9 for SGD.

| Dataset | optimizer | lr | scheduler | batch size | weight decay | epochs |
|---|---|---|---|---|---|---|
| Waterbirds | SGD | 1e-3 | Constant | 32 | 1e-3 | 100 |
| CelebA | SGD | 1e-3 | Constant | 128 | 1e-4 | 50 |
| MultiNLI | AdamW | 2e-5 | Constant | 32 | 1e-4 | 4 |
| CivilComments | AdamW | 2e-5 | Constant | 32 | 1e-4 | 4 |

Table 3: Hyperparameters for sample weight updates (outer loop). For the step learning rate scheduler (StepLR), we simply decay the learning rate by a factor of 10 every 30 steps.

| Dataset | optimizer | lr | scheduler | grad clip | $\tau$ | steps |
|---|---|---|---|---|---|---|
| Waterbirds | GD | 1 | StepLR | 1 | 0.1 | 100 |
| CelebA | GD | 1 | StepLR | 1e-2 | 0.01 | 100 |
| MultiNLI | GD | 0.1 | StepLR | 1e-2 | 1 | 50 |
| CivilComments | GD | 1 | StepLR | 1e-1 | 0.1 | 50 |

Table 4: Hyperparameters for last-layer retraining (inner loop)

| Dataset | optimizer | lr | scheduler | batch size | weight decay | line search |
|---|---|---|---|---|---|---|
| Waterbirds | L-BFGS | 1e-4 | Constant | full | 1e-1 | Strong Wolfe |
| CelebA | L-BFGS | 1e-4 | Constant | full | 1e-2 | Strong Wolfe |
| MultiNLI | L-BFGS | 1e-4 | Constant | full | 1e-2 | Strong Wolfe |
| CivilComments | L-BFGS | 1e-1 | Constant | full | 1e-3 | Strong Wolfe |

### B.3 IMPLEMENTATION DETAILS

Let $d$ denote the dimension of $\theta$. Let the size of the held-out training set be $n$ here. To efficiently compute the influence scores of the samples for last-layer retraining, we adopt the following strategy:

1. Calculate the Hessian inverse $H_{\hat{\theta}^*,w}^{-1}$ of the weighted objective w.r.t. $\theta = \hat{\theta}^*$. The Hessian has a dimension of $d \times d$.

2. Calculate the *per-sample* gradient of the *unweighted* objective w.r.t. $\theta = \hat{\theta}^*$ for the *training* set. This results in a $n \times d$ matrix.

3. Calculate the *per-group* gradient of the *unweighted* objective w.r.t. $\theta = \hat{\theta}^*$ for the *target* set. This results in a $m \times d$ matrix, where $m$ is the number of groups.

By combining these three matrices, we obtain the group-wise influence scores as a $n \times m$ matrix, which can then be aggregated as in Algorithm 1 and become an $n \times 1$ gradient vector.

## B.4 Running Time Comparison

**Hardware.** For all experiments, we run on machines with NVIDIA RTX 3080 (10GB) / RTX A5000 (24GB) GPU and AMD EPYC 7543 CPU.

**Running time.** We record the detailed running time for both the inner and the outer loop for different datasets in Table 5. The results are obtained from running a single job on one NVIDIA RTX 3080 GPU with two AMD EPYC 7543 CPUs.

Table 5: Running time for `GSR`

| Dataset | Dataset stats | | | Running time (s) | | |
|---|---|---|---|---|---|---|
| | Held-out size | Target size | # of steps | Outer loop | Inner loop | Total |
| Waterbirds | 479 | 600 | 100 | 2.71 | 0.16 | 285 |
| CelebA | 16277 | 9934 | 100 | 4.03 | 0.52 | 451 |
| MultiNLI | 20617 | 46231 | 50 | 7.24 | 0.53 | 381 |
| CivilComments | 26903 | 22590 | 50 | 8.21 | 0.7 | 882 |

**Scalability.** As we used `Pytorch` for our implementation and the last-layer retraining has very few parameters, the per-sample gradient and the Hessian inverse calculation involves high CPU usage, even though the tensors are on GPU. When the same processes are running in parallel, the performance sometimes can be CPU-bound instead of GPU-bound. Thus, the running time is likely to increase when the CPU-usage is high.

## B.5 Detailed Results

In addition to the worst-group accuracy shown in the main paper, we also report the mean accuracy for these datasets in Table 6 for reference. We would like to highlight that the worst-group accuracy gain is usually at the expense of the mean accuracy. This implies that trade-offs generally need to be made for better robustness to subpopulation shifts, since the mean accuracy corresponds to an estimation of the model's performance when there is no distribution shift.

## C Understanding the Role of the Inverse Hessian

Koh & Liang (2017) analyzed that the inverse Hessian measures the robustness of the resultant model to upweighting the training data. As Hessian is a symmetric matrix, according to the Spectral Theorem, it guarantees the existence of an orthonormal basis of the eigenvectors. Hence, the gradient can be written as a linear combination of eigenvectors. As the influence function utilizes the inverse Hessian, which has the reciprocal of the eigenvalues of the Hessian, the influence score has a *larger* magnitude when the gradient of the training and test points are similar and along the direction of the eigenvectors with *smaller* magnitudes.

Having smaller eigenvalues can be interpreted as the first-order gradient changing relatively slowly. From the optimization perspective (Newton's method), this usually means we are allowed to take larger gradient steps when the trajectory has less variation with a smoother curvature. On the other hand, the allowed step size becomes smaller when the curvature changes more quickly, indicating a more complex loss landscape.

If the Hessian is dropped in between the inner product of the gradient as in Equation 2 (used by MAPLE), then we fail to account for the variations of step sizes in different directions specified by the eigenvectors. Thus, the outer loop gradient becomes incorrect and the optimization will be less reliable.

## D Visualization of the Spurious Correlations

We show an illustration of the spurious correlations for the Waterbirds dataset in Figure 5.

Table 6: **Performance comparison in detail.** We report the worst-group accuracy on four benchmark datasets. For the group label column, ✗ indicates no group label $g$ is used; ✓ indicates $g$ is required for training or validation; ✓̇ indicates $g$ is partially used; ✓̃ indicates a proportion of $g$ from the validation set is re-purposed beyond hyperparameter tuning, such as training sample weights or model parameters. The row sections are ranked in descending order of the total amount of group label information required.

| Method | Group Label Train / Val | Waterbirds Worst(%) | Waterbirds Mean(%) | CelebA Worst(%) | CelebA Mean(%) | MultiNLI Worst(%) | MultiNLI Mean(%) | CivilComments Worst(%) | CivilComments Mean(%) |
|---|---|---|---|---|---|---|---|---|---|
| RWG | ✓/ ✓ | $87.6_{\pm1.6}$ | — | $84.3_{\pm1.8}$ | — | $69.6_{\pm1.0}$ | — | $72.0_{\pm1.9}$ | — |
| Group DRO | ✓/ ✓ | $91.4_{\pm1.1}$ | $93.5_{\pm0.3}$ | $88.9_{\pm2.3}$ | $92.9_{\pm0.2}$ | $77.7_{\pm1.4}$ | $81.4_{\pm0.1}$ | $70.0_{\pm2.0}$ | $89.9_{\pm0.5}$ |
| JTT | ✗/ ✓ | 86.7 | 93.3 | 81.1 | 88.0 | 72.6 | 78.6 | 69.3 | 91.1 |
| CnC | ✗/ ✓ | $88.5_{\pm0.3}$ | $90.9_{\pm0.1}$ | $88.8_{\pm0.9}$ | $89.9_{\pm0.5}$ | — | — | $68.9_{\pm2.1}$ | $81.7_{\pm0.5}$ |
| SSA | ✗/ ✓̃ | $89.0_{\pm0.6}$ | $92.2_{\pm0.9}$ | $\mathbf{89.8}_{\pm1.3}$ | $92.8_{\pm0.1}$ | $76.6_{\pm0.7}$ | $79.9_{\pm0.9}$ | $69.9_{\pm2.0}$ | $88.2_{\pm2.0}$ |
| MAPLE | ✗/ ✓̃ | 91.7 | 92.9 | 88.0 | 89.0 | 72.7 | 77.2 | 64.1 | 89.7 |
| DFR | ✗/ ✓̃ | $\mathbf{92.9}_{\pm0.2}$ | $94.2_{\pm0.4}$ | $88.3_{\pm1.1}$ | $91.3_{\pm0.3}$ | $74.7_{\pm0.7}$ | $82.1_{\pm0.2}$ | $70.1_{\pm0.8}$ | $87.2_{\pm0.3}$ |
| GSR-HF | ✗/ ✓̃ | $87.5_{\pm0.1}$ | $97.7_{\pm0.0}$ | $86.3_{\pm0.4}$ | $91.5_{\pm0.0}$ | $74.7_{\pm0.0}$ | $80.6_{\pm0.0}$ | $68.9_{\pm0.2}$ | $89.1_{\pm0.0}$ |
| GSR | ✗/ ✓̃ | $\mathbf{92.9}_{\pm0.0}$ | $94.9_{\pm0.1}$ | $87.0_{\pm0.4}$ | $90.9_{\pm0.0}$ | $\mathbf{78.5}_{\pm0.3}$ | $79.8_{\pm0.0}$ | $\mathbf{71.7}_{\pm0.6}$ | $85.9_{\pm0.4}$ |
| ERM | ✗/ ✗ | $74.9_{\pm2.4}$ | $98.1_{\pm0.1}$ | $46.9_{\pm2.8}$ | $95.3_{\pm0.0}$ | $65.9_{\pm0.3}$ | $82.8_{\pm0.1}$ | $55.6_{\pm0.6}$ | $92.1_{\pm0.1}$ |
| ES SELF | ✗/ ✗ | $\mathbf{93.0}_{\pm0.3}$ | $94.0_{\pm1.7}$ | $83.9_{\pm0.9}$ | $91.7_{\pm0.4}$ | $70.7_{\pm2.5}$ | $81.2_{\pm0.7}$ | — | — |

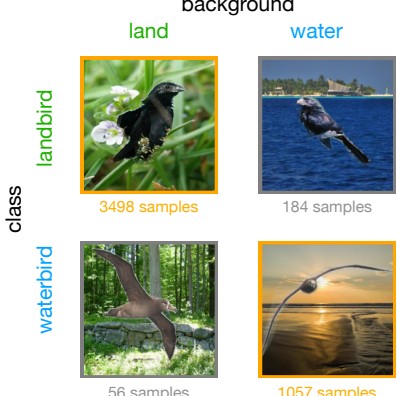

Figure 5: An illustration of the Waterbirds dataset. A spurious correlation exists between the bird class and the background. The majority groups are highlighted in orange. The minority groups are in gray.

# E  WHAT IF WE CAN RETRAIN THE FULL NETWORK WITH GSR?

In this section, we use a simple setup to explore the potential of retraining the full network with GSR, when the compute budget allows.

**Setup.** We conduct an ablation study on a small binary classification task customized from the ColoredMNIST, where the digits are spuriously correlated with color. We use a tiny Convolutional Neural Network (CNN) with 3 layers and about 5000 parameters.

**Dataset.** *ColoredMNIST-S*. We modify the original ColoredMNIST dataset (Arjovsky et al., 2019) by first taking only the first 6 digits out of all 10 digits, then assigning $y < 3$ as 0 and $y \geq 3$ as 1. The training, validation, and test splits have 10000, 2000, and 27375 instances, respectively. The spurious correlation $\gamma$ is created by associating most of the instances in class 0 with red color and class 1 with green color. We set $\gamma = 0.1$ for the training split and set $\gamma = 0.5$ for the validation and test splits. we do not introduce any label corruption. This setting is considered much simpler than the original ColoredMNIST, primarily because we are studying with limited network capacity with a tiny CNN, which could not fit the original 10-digit classification with more than 90% accuracy.

**Baselines.** We compare GSR, which is by default trained with last-layer retraining, with the full-parameter retraining variants. GSR-HF discards the Hessian, which is equivalent to MAPLE+LLR. GSR-F-EH retrains the entire network with the influence function and no approximation on the

Table 7: Performance comparison on ColoredMNIST-S Dataset with TinyCNN.

| Method | ColoredMNIST-S | | Wall-Clock Time |
|---|---|---|---|
| | Worst(%) | Mean(%) | |
| GSR-F-EH | **94.1** | 96.0 | 400 |
| GSR-F-AH | 80.2 | 97.6 | 45 |
| GSR-F-HF | 91.9 | 96.9 | 42 |
| GSR | 91.2 | 95.4 | 4+0.8 |
| GSR-HF | 91.1 | 96.2 | 4+0.7 |
| ERM | 81.2 | 97.7 | 4 |

Table 8: Running time and memory consumption of full-parameter retraining on Waterbirds dataset with ResNet-50.

| Approach | Time (One-iteration) | VRAM Consumption |
|---|---|---|
| Explicit Hessian | >18h | 2PB |
| Conjugate Gradients | 18h | 9GB |
| LiSSA | 3.6h | 9GB |

Hessian. We use Conjugate Gradients to calculate the inverse Hessian-vector product (iHVP) instead of explicitly calculating the Hessian to save the memory. `GSR-F-AH` approximates the Hessian using LiSSA (Agarwal et al., 2017) with depth=1000 and repeat=1. This approximation calculates the iHVP with 1000 samples from the training set. `GSR-F-HF` which discards the Hessian, and this uses the sample gradients as MAPLE.

The results are shown in Table 7. we can observe that. Full-parameter retraining with Hessian (i.e., `GSR-F-EH`) is advantageous over all other approaches. However, approximating the Hessian with low quality might hurt the performance, since `GSR-F-AH` underperforms other baselines. When using a small neural network model, last-layer retraining (LLR) can experience slight performance degradation compared to retraining the whole model on the entire dataset (since we have the capacity to do so), but it is still significantly better than ERM. Overall, these results suggest that we should consider full-parameter retraining when we have the capacity, and LLR is an efficient and effective approximation to it.

### E.1 CHALLENGES OF SCALING UP FULL-PARAMETER RETRAINING WITH GSR

When scaling up full-parameter retraining to more realistic datasets with larger networks, the computational challenges for accurately calculating the sample weight gradients are indeed the major obstacle here because of the Hessian. Let $n$ be the number of data points and $p$ be the number of parameters. Directly calculating the Hessian for the whole network is both computationally expensive $O(np^2 + p^3)$ and memory bottlenecked $O(p^2)$. In particular, the Hessian of a simple neural network with 1M parameters takes 4TB of memory, and ResNet-50 has 23M trainable parameters. Alternatively, we can calculate the inverse Hessian-vector product (iHVP) to reduce the memory requirements to $O(p)$ and the time complexity to $O(np^2)$, but it is still time-consuming, unless we apply further approximations. Table 8 shows the computational challenges of performing full-parameter retraining with GSR on the Waterbirds dataset.

It will be extremely costly to perform the experiments that optimize the sample weights for a few iterations (e.g., 50 iterations, as done with LLR). Besides, as shown in the previous ablation study, training the full network with crudely approximated Hessian (using LiSSA) causes inaccurate gradient estimation and may even hurt the performance. We believe these technical challenges require extensive exploration in future works and are beyond the scope of this paper.

## F COMPARISON WITH MAPLE

Although the difference seems to be only about the Hessian, `GSR` and MAPLE are based on two distinct ways to solve the bilevel optimization problem.

Table 9: Performance comparison on MetaShift.

| Method | MetaShift | |
|---|---|---|
| | Worst(%) | Mean(%) |
| GSR | $\mathbf{78.2}_{\pm 4.0}$ | $87.9_{\pm 4.1}$ |
| DFR | $71.4_{\pm 1.6}$ | $89.7_{\pm 0.2}$ |
| ERM | $69.2$ | $90.3$ |

Table 10: Performance comparison on modified datasets where the spurious correlations are extremely strong so that the original minority groups no longer exist.

| | Waterbirds- | MNLI- |
|---|---|---|
| GSR | $87.8_{\pm 0.1}(-5.1)$ | $\mathbf{74.2}_{\pm 0.1}(-4.3)$ |
| DFR | $\mathbf{89.1}_{\pm 0.5}(-3.8)$ | $66.9_{\pm 0.2}(-7.8)$ |
| ERM | $32.6(-42.3)$ | $9.0(-56.9)$ |

1. MAPLE directly approximates backpropagating through the training trajectory by one-step truncation, but GSR uses implicit differentiation that results in the Hessian. This calculation is *exact* when the objective is strongly convex.

2. GSR also synergies with last-layer retraining, which was not considered by MAPLE.

3. We use an adaptive aggregation of the influence scores from the groups as the gradient, but MAPLE does not explicitly design group-level aggregation for subpopulation shifts.

4. MAPLE additionally uses coreset selection by training on a subset of all the training data in each step to speed up the training, while GSR utilizes the full training data without approximation.

## G  ADDITIONAL DATASET FOR SUBPOPULATION SHIFT

To further demonstrate the advantage of GSR, we include an additional experiment with the *MetaShift* (Liang & Zou, 2022) image classification dataset. We use the two-class version implemented by Yang et al. (2023), where the spurious correlation exists between the class and the background. In particular, cats are mostly indoors and dogs are mostly outdoors. We primarily compare with DFR, which uses the same amount of group labels and the same ERM feature extractor as ours. The ERM model is trained for 100 epochs without early stopping or hyperparameter tuning. GSR outperforms DFR convincingly in this case as shown in Table 9.

## H  ADDITIONAL EMPIRICAL STUDY

### H.1  WHAT IF THE ERM-LEARNED REPRESENTATIONS ARE INSUFFICIENT?

To simulate the situation where ERM fails to learn sufficiently good representations, we deliberately remove the minority groups from the training data that are used for representation learning, then followed by GSR or DFR with LLR. We experimented with two modified datasets: Waterbirds- and MNLI-, where the spurious correlations are more severe ($\approx 1$). We only compare with DFR, because other baselines (such as JTT, Group DRO) will not work since they cannot use the minority-group data from training. We report the change in performance in Table 10 (the number in brackets indicates the performance change compared to learning the representations with the minority groups). Clearly, when the representation quality degrades, the performance drops across different methods. In particular, ERM performs much worse than random guesses for the worst group. Nonetheless, some useful representations are still learned and can be efficiently utilized by last-layer retraining guided by some minority data.

