# OpenReview forum: "Group-robust Sample Reweighting for Subpopulation Shifts via Influence Functions"
_ICLR.cc/2025/Conference — ICLR 2025 Poster_

### Official Review · Reviewer_oDPM · 2024-10-30

**Soundness:** 2
**Presentation:** 3
**Contribution:** 3
**Rating:** 6
**Confidence:** 3

**Summary:**

This paper proposes a method to mitigate spurious correlations and improve group robustness. The task is formulated as a bilevel optimization problem, which the authors solve via implicit differentiation. They employ Last-Layer Retraining (LLR) to ensure that the optimization problem meets the conditions required for implicit differentiation. Experiments on several datasets demonstrate the effectiveness of the proposed method.

**Strengths:**

- This paper utilizes implicit differentiation to calculate the gradient with respect to the sample weights and applies last-layer retraining to ensure that the risk function satisfies the conditions of convexity and continuous differentiability.

- Overall, the paper is written and organized well.

- The authors not only conducted experiments on benchmark datasets but also evaluated the performance of the proposed method in the presence of label noise. The proposed method demonstrated effectiveness in identifying noisy data.

**Weaknesses:**

- There appears to be an issue with the proof in Appendix A.2. By examining the first and third lines together, we derive $\nabla_{w_i} \theta_T \approx \nabla_{w_i} \left( \theta_T - \eta \nabla_\theta \hat{\mathcal{R}}(\mathcal{D}^{\text{tr}}; w, \theta_T) \right)$ which then leads to $0 \approx - \eta \nabla_\theta \hat{\mathcal{R}}(\mathcal{D}^{\text{tr}}; w, \theta_T)$, which seems unreasonable.
Furthermore, in the transition from the third to the fourth line, it is unclear why $\nabla_{w_i} \theta_T$ was eliminated. Was it because $\nabla_{w_i} \theta_T = 0$? If that is the case, the proof in Appendix A.2 could directly conclude that $\nabla_{w_i} \hat{\theta}^* \approx \nabla_{w_i} \theta_T \approx 0$. It might be worthwhile for the authors to review this part of the proof carefully.
I do not believe this issue impacts the main contributions of the paper, but the authors may need to consider revising the related explanations in the paper that are based on this proof.

- In Appendix A.3, it is assumed that when $\nabla_{\theta} \hat{\mathcal{R}}(\mathcal{D}^{tr}; w, \hat{\theta}^*) = 0$, it follows that $\frac{\mathrm{d} \nabla_{\theta} \hat{\mathcal{R}}(\mathcal{D}^{tr}; w, \hat{\theta}^*)}{\mathrm{d} w} = 0$. Could the authors elaborate on why the gradient with respect to $\theta$ being zero implies that its derivative with respect to $w$ is also zero in this context?

**Questions:**

see weaknesses.

---

> ### Author Response · Authors · 2024-11-21
> **Author Response**
>
> We thank the reviewer for carefully checking the proof and recognizing the contribution and effectiveness of our method in improving group robustness and noise robustness. The detailed responses are as follows:
>
> **Summary:**
>
> - We have addressed a notational error in the proof.
> - We provided more explanations for the derivations of our sample gradient update using implicit differentiation.
>
> **Weaknesses:**
>
> **W1:**
>
> > There appears to be an issue with the proof in Appendix A.2 …  It might be worthwhile for the authors to review this part of the proof carefully. I do not believe this issue impacts the main contributions of the paper, but the authors may need to consider revising the related explanations in the paper that are based on this proof.
> >
>
> Thank you for spotting this out. We apologize for one notational mistake on the third line that is misleading, but this does not impact the correctness of the final result. In particular, the first $\theta_{T}$ in the third line should be changed to $\theta_{T-1}$, resulting in $\nabla_{w_i} \left(\theta_{T-1} - \eta \nabla_{\theta}\hat{\mathcal{R}}(D^{\mathrm{tr}}; w, \theta_{T})\right)$ . Although $\theta_T \approx \theta_{T-1}$ in value, we should only use it for $\nabla_{w_i} \nabla_{\theta}\hat{\mathcal{R}}(\mathcal{D}^{\mathrm{tr}}; w, \theta_{T}) \approx \nabla_{w_i} \nabla_{\theta}\hat{\mathcal{R}}(\mathcal{D}^{\mathrm{tr}}; w, \theta_{T-1})$ instead of $\nabla_{w_i} \theta_{T-1} \approx \nabla_{w_i} \theta_{T}$. This is because according to MAPLE’s one-step truncated backpropagation, the gradients and updates before $\theta_{T-1}$ are truncated, so that $\theta_{T-1}$ is a “constant” and is **independent** of $w$, but $\theta_T$ remains as a function of $w$. Thus, it is true that $\nabla_{w_i} \theta_{T-1} = 0$, but not necessarily for $\nabla_{w_i} \theta_{T}$. We have revised the derivation steps with more explanations in Appendix A.2.
>
> ---
>
> **W2:**
>
> > In Appendix A.3, it is assumed that when $\nabla_{\theta} \hat{\mathcal{R}}(\mathcal{D}^\mathrm{tr}; w, \hat{\theta}^*) = 0$, it follows that $\frac{\mathrm{d} \nabla_{\theta} \hat{\mathcal{R}}(\mathcal{D}^\mathrm{tr}; w, \hat{\theta}^*)}{\mathrm{d} w} = 0$. Could the authors elaborate on why the gradient with respect to $\theta$ being zero implies that its derivative with respect to $w$ is also zero in this context?
> >
>
> We can begin by thinking $F(w) =\nabla_{\theta} \hat{\mathcal{R}}(\mathcal{D}^\mathrm{tr}; w, \hat{\theta}^*)$ as a function of $w$, i.e., the function to be differentiated w.r.t. $w$ is the “gradient w.r.t. $\theta$” instead of “$\hat{\mathcal{R}}$”. By differentiating w.r.t. $w$ on both sides of the equation  $\nabla_{\theta} \hat{\mathcal{R}}(\mathcal{D}^\mathrm{tr}; w, \hat{\theta}^*) = 0$, we have $\frac{\mathrm{d} \nabla_{\theta} \hat{\mathcal{R}}(\mathcal{D}^\mathrm{tr}; w, \hat{\theta}^*)}{\mathrm{d} w} = \frac{\mathrm{d}0}{\mathrm{d} w} = 0$. This is usually the first step of implicit differentiation. For clarification, the result from implicit differentiation is the same as: for a constant function $f(x) = C,~ f'(x) = 0$ where $x$ is a variable. We have added this step in A.3 accordingly.
>
> ---
>
> We hope these clarifications have addressed your concerns about our theoretical soundness and will further improve your opinion of our work.

---

> ### Author Response · Authors · 2024-11-25
> **Gentle Reminder**
>
> Dear Reviewer oDPM, we sincerely appreciate your thorough review of our work. We would like to confirm if our clarifications above have resolved your concerns and we are happy to address any remaining doubts.

---

> > ### Comment · Reviewer_oDPM · 2024-11-26
> >
> > Thank you for the comprehensive response to my review. Your explanation has successfully resolved my concerns, and as a result, I am pleased to increase my review score.

---

> > > ### Author Response · Authors · 2024-11-27
> > >
> > > Thank you sincerely for increasing the score and recognizing the improved soundness of our revision!

---

### Official Review · Reviewer_xdjG · 2024-10-30

**Soundness:** 3
**Presentation:** 2
**Contribution:** 2
**Rating:** 6
**Confidence:** 3

**Summary:**

This paper introduces a novel approach called Group-robust Sample Reweighting (GSR) aimed at mitigating the issue of spurious correlations, particularly under subpopulation shifts. The paper proposes a two-stage strategy that utilizes group-labeled data more efficiently by leveraging it to reweight group-unlabeled data. The key contribution of the GSR method is its use of last-layer retraining (LLR), which simplifies the bilevel optimization problem by focusing only on the last layer of the model. Through iterative reweighting of the group-unlabeled data based on influence functions and adaptive aggregation of sample gradients, GSR aims to improve the worst-group accuracy.

**Strengths:**

1. The proposed method is built on a solid theoretical foundation, particularly leveraging bilevel optimization and influence functions, which enhances its robustness and credibility.

2. By focusing on the last layer, the method simplifies the optimization process, making it computationally efficient.

3. The paper provides extensive experimental results across a variety of benchmarks and other understanding experiments.

**Weaknesses:**

1. In Section 2, it would help to be more explicit about how the proposed approach differs from previous methods. Right now, it feels like Section 3 is mostly about improving an already established objective (MAPLE). Adding the inverse Hessian in the optimization might make things more complicated than before. It’s not clear from the paper whether this extra complexity is worth the performance gains.

2. It’s not obvious how much of the performance boost comes from last-layer retraining (LLR) versus the optimization changes over MAPLE. It would be useful to see a comparison where MAPLE is combined with LLR, so we can better understand where the gains are really coming from.

3. The claim that “over-reliance on spurious correlations mainly occurs in the last layer” feels a bit questionable. Prior research indicates that neural networks can develop biases throughout the entire network, not just in the final layer, e.g, [1]. It would be helpful to either provide more evidence or explain this assumption more clearly.

4. Table 1 is missing some crucial results, especially for MAPLE, which is an important baseline here. The paper mentions that it couldn’t get the results from the original paper, but why not try reproducing them? Was there a specific issue? It feels like an incomplete comparison without those numbers.

[1]What neural networks memorize and why: Discovering the long tail via influence estimation

**Questions:**

Please see above

---

> ### Author Response · Authors · 2024-11-21
> **Author Response (Part 1)**
>
> We appreciate the reviewer for the insightful comments and for recognizing our solidity in the theoretical foundation and the comprehensiveness of our experimental study. Our detailed responses to the comments are as follows:
>
> **Summary:**
>
> - We have clarified the differences between our approach and MAPLE and how our GSR-HF baseline is essentially MAPLE with last-layer retraining.
> - We have revised the statement that justifies the use of last-layer retraining and provided additional empirical support.
> - We have reimplemented MAPLE and filled its missing entries in Tables 1 and 6.
>
> **Weaknesses:**
>
> **W1:**
>
> > In Section 2, it would help to be more explicit about how the proposed approach differs from previous methods. Right now, it feels like Section 3 is mostly about improving an already established objective (MAPLE). Adding the inverse Hessian in the optimization might make things more complicated than before. It’s not clear from the paper whether this extra complexity is worth the performance gains.
> >
>
> We have revised the paper by adding a clarification about the differences between our approach and MAPLE in Section 1 (lines 77-79) with more details in Appendix F. Our approach differs from MAPLE in the following ways:
>
> - GSR and MAPLE are two **distinct** ways to solve the bilevel optimization problem.
>     - MAPLE directly approximates the outer loop gradient by a one-step truncation. The target of approximation is “**backpropagating through the training trajectory**”.
>     - GSR calculates the outer loop gradient via “implicit differentiation” that requires the Hessian. This calculation is *exact* when the objective is strongly convex and **avoids the need to unroll the training trajectory**.
> - GSR also synergies with last-layer retraining (LLR), which was **not** considered by MAPLE.
> - GSR uses an adaptive aggregation of the influence scores from the groups as the gradient (discussed in lines 208-220), but MAPLE does **not** explicitly design group-level aggregation for subpopulation shifts.
>
> In addition, the gradient inner product for MAPLE derived in Equation (3) is in fact by our effort and was **not** discussed in MAPLE’s original paper. It helps the flow of our paper by providing a clear and deeper understanding of our differences.
>
> Finally, our approach does not add computational complexities due to the use of LLR. Unlike MAPLE, we do not need re-differentiation or re-training of the full network, either. Hence, our GSR is significantly simpler and faster in practice than MAPLE.
>
> ---
>
> **W2:**
>
> > It’s not obvious how much of the performance boost comes from last-layer retraining (LLR) versus the optimization changes over MAPLE. It would be useful to see a comparison where MAPLE is combined with LLR, so we can better understand where the gains are really coming from.
> >
>
> The Hessian-free version of our approach, GSR-HF, which uses Equation (3) for sample reweighting, is in fact the **same as MAPLE+LLR**, as discussed in lines 357-361. It can be seen from Table 1 that GSR-HF performs comparably to MAPLE (~79% accuracy on average). Hence, adding LLR to MAPLE does **not** directly result in performance boosts.
>
> Rather, requiring the Hessian in our GSR ensures accurate gradient estimation and is more helpful to the gradient descent algorithm that we use to optimize the sample weights. Notably, GSR ***significantly outperforms*** GSR-HF, justifying the importance of the Hessian.

---

> ### Author Response · Authors · 2024-11-21
> **Author Response (Part 2)**
>
> **W3:**
>
> > The claim that “over-reliance on spurious correlations mainly occurs in the last layer” feels a bit questionable. Prior research indicates that neural networks can develop biases throughout the entire network, not just in the final layer, e.g, [1]. It would be helpful to either provide more evidence or explain this assumption more clearly.
> >
>
> Thank you for pointing it out. We apologize for causing the confusion and would like to clarify that we meant that “**over-reliance on spurious correlations can be significantly reduced by simply fine-tuning the last linear classification layer**”. We have revised the main paper accordingly (lines 236-238). This claim is a hypothesis that was proposed and studied by prior research in the area of distribution shifts [2,3,4], where LLR closely matches the performance of more sophisticated full-parameter retraining approaches, even though it uses the representations from a poor ERM-trained model. In addition, although our approach focuses on LLR, it demonstrates better robustness to subpopulation shifts by outperforming many existing baselines that train/retrain on the **entire network**, such as Group DRO, JTT, CnC, SSA, MAPLE, etc. This empirical efficacy also further validates the hypothesis.
>
> To further illustrate, we empirically construct a case where **minority groups are completely missing** from the training set (spurious correlation $\approx$ 1). We first train an ERM model, which has poor generalization to the minority groups,  then apply LLR methods on top of the ERM representations and report their worst-group accuracy. The table below shows the results for Waterbirds- and MNLI- datasets, where the number in brackets indicates the performance degradation compared to Table 1. We can observe that LLR methods are still able to mitigate spurious correlations effectively.
>
> |  | Waterbirds- | MNLI- |
> | --- | --- | --- |
> | GSR (with LLR) | $87.8_{\pm 0.1} (-5.1)$ | $74.2_{\pm 0.1} (-4.3)$ |
> | DFR (with LLR) [2] | $89.1_{\pm 0.5} (-3.8)$ | $66.9_{\pm 0.2} (-7.8)$ |
> | ERM (without LLR) | $32.6 (-42.3)$ | $9.0 (-56.9)$ |
>
> Finally, as discussed in Section 7 (lines 536-539), we believe that it is a promising direction for the future work to explore whether retraining the full network with reweighted samples **at scale** can improve the representation learning and the robustness.
>
> ---
>
> **W4:**
>
> > Table 1 is missing some crucial results, especially for MAPLE, which is an important baseline here. The paper mentions that it couldn’t get the results from the original paper, but why not try reproducing them? Was there a specific issue? It feels like an incomplete comparison without those numbers.
> >
>
> We did not provide the results initially because:
>
> - MAPLE only partially open-sourced their codebase for synthetic MNIST data, which is **not** directly usable on the standard benchmark datasets such as Waterbirds and CelebA.
> - MAPLE’s original implementation is based on the second-order derivative (even though we derive that it is the first-order derivative), which causes **incompatibility** with the latest Transformer models for NLP datasets (MNLI, CivilComments).
> - MAPLE necessitates the training of the model with SGD, which is also **incompatible** with other optimizers such as AdamW, which is often required by Transformer models.
>
> In addition, MAPLE is **computationally challenging** to scale up to NLP datasets with bigger models.  MAPLE reported a training time of 6.74 hours for Waterbirds, which is the smallest dataset with only 5k instances. The other three datasets all contain 160k or more instances, which becomes practically infeasible to train with the original repository.
>
> To address these challenges, we reimplemented the sample reweighting strategy of MAPLE with our derived formulation and improved efficiency, and integrated it into our codebase. Then we run our version of MAPLE on the benchmarks as shown in the table below (which is also updated in Table 1 of the main paper):
>
> |  | Waterbirds | CelebA | MNLI | CivilComments |
> | --- | --- | --- | --- | --- |
> | MAPLE (Our Impl) | 84.0 | 87.7 | 72.7  | 64.1  |
> | MAPLE (Original) | 91.7 | 88.0 | - | - |
>
> To the best of our ability, we fill the results for MNLI and CivilComments after closely matching the original result for MAPLE in CelebA. Unfortunately, we are unable to exactly reproduce the result for Waterbirds as we observe the training does not converge smoothly.
>
> Finally, we would like to highlight that the previous state-of-the-art approach on these benchmarks was DFR instead of MAPLE, which we also outperformed overall.
>
> ---
>
> We hope that the clarifications and additional experiments included in the revision have addressed your concerns and will enhance your evaluation of our work.

---

> ### Author Response · Authors · 2024-11-21
> **Author Response (Part 3)**
>
> **References**
>
> [1] Vitaly Feldman and Chiyuan Zhang. "What neural networks memorize and why: Discovering the long tail via influence estimation." In Advances in Neural Information Processing Systems, 2020.
>
> [2] Polina Kirichenko, Pavel Izmailov, and Andrew Gordon Wilson. Last layer re-training is sufficient for robustness to spurious correlations. arXiv preprint arXiv:2204.02937, 2022.
>
> [3] Pavel Izmailov, Polina Kirichenko, Nate Gruver, and Andrew G Wilson. On feature learning in the presence of spurious correlations. In Advances in Neural Information Processing Systems, 2022.
>
> [4] Tyler LaBonte, Vidya Muthukumar, and Abhishek Kumar. Towards last-layer retraining for group robustness with fewer annotations. In Advances in Neural Information Processing Systems, 2024.

---

> ### Author Response · Authors · 2024-11-25
> **Gentle Reminder**
>
> Dear Reviewer xdjG, we would like to extend our thanks for your thoughtful review. If you have any remaining questions or concerns not addressed in our rebuttal, we would be glad to discuss them further.

---

### Official Review · Reviewer_kehk · 2024-10-31

**Soundness:** 3
**Presentation:** 2
**Contribution:** 2
**Rating:** 6
**Confidence:** 3

**Summary:**

This paper studies the challenge of improving machine learning model robustness to subpopulation shifts, where models may perform inconsistently across different subgroups within data distributions. Traditional methods typically rely on group-labeled data to tune models and minimize worst-case loss across groups, but these approaches are limited by the high cost of obtaining substantial, high-quality labels. The authors propose an alternative framework, Group-robust Sample Reweighting (GSR), which enhances group label efficiency by utilizing group-labeled data as a target to optimize weights for group-unlabeled data. The two-stage GSR approach first learns representations from group-unlabeled data and then fine-tunes the model’s last layer by iteratively retraining on reweighted data. GSR demonstrates superior performance on benchmark datasets.

**Strengths:**

1. The paper highlights that previous methods estimate weight gradients through approximations, but they neither account for the curvature of the loss landscape with respect to the current model nor fully capture the training dynamics. The authors’ proposed method addresses these issues under the assumptions that the inner objective is unconstrained, strongly convex, and twice continuously differentiable.

2. The experiments in this paper are thorough, and the results are promising. Across multiple datasets, the proposed approach outperforms various baseline methods, achieving state-of-the-art performance.

**Weaknesses:**

1. One limitation of this paper is that the proposed method relies on the inner objective being unconstrained, strongly convex, and twice continuously differentiable—assumptions whose strength is not fully clarified. The authors address these requirements by fixing the feature layers during training and only updating the linear classifier layer. However, further explanation is needed to clarify why training only the classifier layer meets these assumptions. Additionally, limiting training to the classifier layer may risk sacrificing overall model performance, which warrants further investigation.

2. The authors' second motivation is somewhat confusing. The statement that "regularization techniques must be applied such that training with weighted objectives leads to meaningfully different solutions than using ERM" may require further clarification. It would be helpful for the authors to elaborate on how this principle is specifically reflected in the design of their method.

3. The authors should conduct additional experiments to examine, without computational constraints, how much performance is lost when training the model with frozen feature layers compared to full-parameter training. This would help determine whether the performance reduction is within an acceptable range.

**Questions:**

See Weaknesses.

---

> ### Author Response · Authors · 2024-11-21
> **Author Response**
>
> We thank the reviewer for providing insightful comments and recognizing the thoroughness of our experiments. Our detailed responses are as follows:
>
> **Summary:**
>
> - We have clarified why last-layer retraining meets the assumptions made in the paper with theoretical justifications.
> - We elaborate on why regularizations are needed for the weighted objective to be meaningfully different than the unweighted objective.
> - We compare full-parameter retraining vs. last-layer retraining with GSR at a manageable scale, then discuss the challenges of extending such comparison to larger scales.
>
> **Weaknesses:**
>
> **W1:**
>
> > **(W1.1)** One limitation of this paper is that the proposed method relies on the inner objective being unconstrained, strongly convex, and twice continuously differentiable—assumptions whose strength is not fully clarified. The authors address these requirements by fixing the feature layers during training and only updating the linear classifier layer. However, further explanation is needed to clarify why training only the classifier layer meets these assumptions.
> >
>
> Using last-layer retraining as the inner objective meets these assumptions because it reduces the problem to **linear classification using the cross-entropy loss**, denoted as $L(x, y;\theta)$, which is unconstrained, twice continuously differentiable, and convex. The key idea of the proof is that we can show the Hessian of $L$ w.r.t $\theta$ exists and is positive semidefinite. Then adding $\ell_2$-regularization with positive strength $\lambda > 0$ makes it **strongly** convex. We have developed detailed proofs and justifications in Appendix A.4 to facilitate the understanding of these concepts and statements. We also added the reference to the proof in Section 4.1 (line 245). We hope these additional materials can help to clarify the concerns.
>
> > **(W1.2)** Additionally, limiting training to the classifier layer may risk sacrificing overall model performance, which warrants further investigation.
> >
>
> We believe this part of the review is the same as **weakness 3**. Please kindly refer to our responses below and in the general response.
>
> ---
>
> **W2:**
>
> > The authors' second motivation is somewhat confusing. The statement that "regularization techniques must be applied such that training with weighted objectives leads to meaningfully different solutions than using ERM" may require further clarification. It would be helpful for the authors to elaborate on how this principle is specifically reflected in the design of their method.
> >
>
> The rationale behind this motivation is that training a weighted objective with overparameterized neural networks **eventually** will converge to the same solution as with unweighted ERM as shown in the previous works [1, 2]. This is because the network has the capacity to reach approximately **0 training loss** on all data points, which diminishes the effect of the training sample weights as all samples are learned equally well. A relevant discussion is in Section 4.1 (lines 247-253). We have revised our presentation in Section 1 (lines 65-68) for better illustration. This principle leads us to use last-layer-retraining with $\ell_2$ regularization, which is one of the ways to **regularize** the model by limiting the expressiveness. This facilitates the model to converge to meaningfully different solutions after training on reweighted samples.
>
> ---
>
> **W3:**
>
> > The authors should conduct additional experiments to examine, without computational constraints, how much performance is lost when training the model with frozen feature layers compared to full-parameter training. This would help determine whether the performance reduction is within an acceptable range.
> >
>
> Firstly, we would like to argue that there may **not** necessarily always be a performance reduction compared to full-parameter retraining. The reason is that in the standard subpopulation benchmarks, our GSR outperforms other methods that require **full-parameter** retraining, including MAPLE, SSA, JTT, CnC, Group DRO, etc.
>
> Secondly, as suggested, we devise an empirical comparison between full-parameter retraining and last-layer retraining at a manageable scale, then analyze the computational challenges of scaling up full-parameter retraining. Please kindly refer to the **General Response** for the details.
>
> ---
>
> We hope that our additional theoretical and empirical analysis has addressed your concerns and will improve your evaluation of our work.
>
> **References**
>
> [1] Runtian Zhai, Chen Dan, Zico Kolter, and Pradeep Ravikumar. Understanding why generalized reweighting does not improve over ERM. arXiv preprint arXiv:2201.12293, 2022.
>
> [2] Shiori Sagawa, Pang Wei Koh, Tatsunori B Hashimoto, and Percy Liang. Distributionally robust neural networks for group shifts: On the importance of regularization for worst-case generalization. arXiv preprint arXiv:1911.08731, 2019.

---

> ### Author Response · Authors · 2024-11-25
> **Gentle Reminder**
>
> Dear Reviewer kehk, thank you once again for your thorough review. We would like to ask if our rebuttal has addressed your concerns. If there are any other questions, we hope to clarify them in time during the discussion period.

---

> > ### Comment · Reviewer_kehk · 2024-11-27
> >
> > The author has addressed my concerns through extensive experimental validation. After reviewing the feedback from other reviewers, I have decided to increase my score to 6.

---

> > > ### Author Response · Authors · 2024-11-27
> > >
> > > Thank you sincerely for the score increase and the recognition of our revision!

---

### Official Review · Reviewer_wcXj · 2024-11-01

**Soundness:** 4
**Presentation:** 2
**Contribution:** 3
**Rating:** 6
**Confidence:** 2

**Summary:**

The paper considers the case when samples can be grouped (not all training samples are necessarily grouped, but all validation samples and target samples must be grouped), and we consider training the prediction model to minimize the maximum groupwise risk (loss) among all groups, namely the "worst-group risk". To train the model under the situation above, the paper considers re-weighting samples during updating the model. Different from an existing method MAPLE, the proposed method considers using the Hessian of the risk when re-weighting. Also, the proposed method introduces the "last-layer retraining" to mitigate spurious correlations and improve group robustness and also to reduce the computational cost.

**Strengths:**

The combination of the sample reweighting and the last-layer retraining to tackle with subpopulation shifts and (consequent) spurious correlations. As far as the reviewer's understanding, (i) a baseline method MAPLE (Zhou et al.; 2022) updates sample weights $w$ for group robustness, but it uses not the Hessian $H_{\hat{\theta}^*,w}$ but only the last-step gradient. (ii) The use of the Hessian raises the computational cost, however, the proposed method focuses on the fact that the last-layer retraining is both costless even we employ the Hessian, and effective for group robustness.

**Weaknesses:**

-   Technical novelty looks not well emphasized. Although the advantage above ("Strengths" section) seems to be effective, the reviewer could find it by carefully reading the paper. This fact is good to be emphasized in a early part of the paper, e.g., in Section 1.
-   The experiments are desired to be more extended. One is to increase the number of datasets (perhaps are there not so many appropriate datasets?), and the other is to qualitative discussions. (How the results are well written, but the discussion on the reasons of the results is desired more.)

**Questions:**

-   The proposed method employs the last-layer retraining, but is it known what types of datasets are not sufficiently learned by the last-layer retraining? If so, comparing performances between methods for such datasets may be interesting.
-   Section 1: It may be better to present an example of spuriously correlated datasets due to groups, e.g., using the "Waterbirds" dataset.
-   Section 3.1: Assumption 3.1 states that, we assume that $\hat{R}({\cal D}^\mathrm{tr}; w,\theta)$ is strongly convex. However, with respect to what variable(s) should we assume it is strongly convex? All of $(w,\theta)$? Please specify.
-   Section 4.1: It states that "Secondly, calculating the Hessian inverse in Equation 6 is no longer overly expensive, because the only free model parameters are in the last layer instead of the entire deep neural network", but how the Hessian inverse can be updated in accordance with the update of $\psi$? It looks a key point of the method, so it is good to be explicitly presented.
-   Section 5.1: The method GSR-HF (Hessian-free) is examined as an ablation study, but how about the ablation study of using the Hessian but not LLR? To compare with MAPLE, this ablation study is also desired. (Perhaps is it too costly?)

---

> ### Author Response · Authors · 2024-11-21
> **Author Response (Part 1)**
>
> We thank you for the insightful comments and for recognizing our technical novelty and the effectiveness of the proposed approach. We have revised the main paper accordingly in **blue-colored** text. Detailed responses are as follows:
>
> **Summary:**
>
> - We have revised the writing and added a visual illustration to improve the presentation by clearer elaboration of the technical novelty, examples of spurious correlation datasets, and mathematical equations.
> - We have included the experimental results on an additional dataset to further demonstrate the effectiveness of our approach.
> - We have performed an ablation study that compares full-parameter retraining with other baselines at a manageable scale and then discussed the challenges of scaling up the study.
>
> **Weaknesses:**
>
> **W1:**
>
> > Technical novelty looks not well emphasized. Although the advantage above ("Strengths" section) seems to be effective, the reviewer could find it by carefully reading the paper. This fact is good to be emphasized in a early part of the paper, e.g., in Section 1.
> >
>
> According to the suggestions, we have updated the introduction section to more clearly highlight the technical novelty (lines 75-79).
>
> ---
>
> **W2:**
>
> > The experiments are desired to be more extended. One is to increase the number of datasets (perhaps are there not so many appropriate datasets?), and the other is to qualitative discussions. (How the results are well written, but the discussion on the reasons of the results is desired more.)
> >
>
> We would like to highlight that the four datasets, spanning both image and text modalities, are the standard benchmark datasets usually used in prior work. We believe that these results provide compelling evidence of the effectiveness of our approach.
>
> Nevertheless, to further strengthen the argument, we include an additional experiment with the ***MetaShift*** [1] image classification dataset. We use the two-class version implemented by [2], where the spurious correlation exists between the class and the background. In particular, cats are mostly indoors and dogs are mostly outdoors. We primarily compare with DFR (the best-performing baseline according to our evaluations), which uses the same number of group labels and the same ERM feature extractor as ours. The ERM model is trained for 100 epochs without early stopping or hyperparameter tuning. GSR (with LLR) **outperforms** DFR convincingly in this case as shown in the table below for the MetaShift dataset:
>
> | **Method** | Worst (%) | Mean(%) |
> | --- | --- | --- |
> | GSR | $78.2_{\pm 4.0}$ | $87.9_{\pm 4.1}$ |
> | DFR | $71.4_{\pm 1.6}$ | $89.7_{\pm 0.2}$ |
> | ERM | $69.2$ | $90.3$ |
>
> These results are updated in Appendix G.
>
> Regarding the discussions of the results, we have expanded our discussion in Section 5.1. We have also presented quantitative and qualitative analysis in Section 5.2 which mainly discusses how GSR is capable of upweighting the minority samples and downweighting the majority samples. In addition, reweighting at the instance level allows more fine-grained control over the sample weights, which offers benefits such as more flexibility and noise robustness. These analyses are meant to explain the advantages of GSR.

---

> ### Author Response · Authors · 2024-11-21
> **Author Response (Part 2)**
>
> **Questions:**
>
> **Q1:**
>
> > The proposed method employs the last-layer retraining, but is it known what types of datasets are not sufficiently learned by the last-layer retraining? If so, comparing performances between methods for such datasets may be interesting.
> >
>
> Prior work has shown that last-layer retraining (LLR) may not be sufficient for domain generalization tasks [2], where the test data contain unseen subpopulations during the training stage (e.g., training on photographic images but the test data contains human sketches). Domain generalization is considered a more difficult setting because the in-domain representations may not be useful out-of-domain. As we focus on mitigating spurious correlations, particularly for subpopulation shifts, we believe a full evaluation for domain generalization is beyond the scope of this work.
>
> To simulate the situation where ERM fails to learn a sufficiently good representation, we deliberately **remove the minority groups** from the training data that are used for representation learning, then followed by GSR or DFR with LLR. We experimented with Waterbirds- and MNLI-, where the spurious correlations are more severe ($\approx$ 1). We only compare with DFR, because other baselines (such as JTT, Group DRO) will not work since they cannot use the minority-group data from training. We report the change in performance in the following table (the number in brackets indicates the performance change compared to learning the representations with the minority groups):
>
> |  | Waterbirds- | MNLI- |
> | --- | --- | --- |
> | GSR | $87.8_{\pm 0.1} (-5.1)$ | $74.2_{\pm 0.1} (-4.3)$ |
> | DFR | $89.1_{\pm 0.5} (-3.8)$ | $66.9_{\pm 0.2} (-7.8)$ |
> | ERM | $32.6 (-42.3)$ | $9.0 (-56.9)$ |
>
> Clearly, when the representation quality degrades, the performance drops across different methods. In particular, ERM performs much worse than random guess for the worst group. **Nonetheless, some useful representations are still learned and can be efficiently utilized by last-layer retraining guided by some minority data.**
>
> ---
>
> **Q2:**
>
> > Section 1: It may be better to present an example of spuriously correlated datasets due to groups, e.g., using the "Waterbirds" dataset.
> >
>
> We appreciate the suggestion and we have added an illustration of the spurious correlation using the Waterbirds dataset in Figure 5, Appendix D. To avoid breaking the line numbers in the main paper, we will move it into Section 1 in future revision.
>
> ---
>
> **Q3:**
>
> > Section 3.1: Assumption 3.1 states that, we assume that $\hat{R}({\cal D}^\mathrm{tr}; w,\theta)$ is strongly convex. However, with respect to what variable(s) should we assume it is strongly convex? All of (w,θ)? Please specify.
> >
>
> Thank you for pointing this out. We should have clarified that all criteria in Assumption 3.1 are for the model parameters  $\theta$. As discussed in line 164-170, the criteria are to ensure the desirable properties on the gradient $\nabla_\theta \hat{R}({\cal D}^\mathrm{tr}; w,\hat{\theta}^*)$ and the Hessian $\nabla_\theta^2 \hat{R}({\cal D}^\mathrm{tr}; w,\hat{\theta}^*)$ with respect to $\theta$ and a fixed $w$. That is not to say we do not require any assumption on $w$, because $\nabla_\theta \hat{R}({\cal D}^\mathrm{tr}; w,\hat{\theta}^*)$ still needs to be differentiable w.r.t. $w$, which is omitted because it is less critical and easily satisfied, as $w$ controls the weighted sum of the instance-level losses. We have revised accordingly in Assumption 3.1.
>
> ---
>
> **Q4:**
>
> > Section 4.1: It states that "Secondly, calculating the Hessian inverse in Equation 6 is no longer overly expensive, because the only free model parameters are in the last layer instead of the entire deep neural network", but how the Hessian inverse can be updated in accordance with the update of $\psi$? It looks a key point of the method, so it is good to be explicitly presented.
> >
>
> At time step $t$, after we train the last-layer classifier with sample weight $w^{(t)}$ and obtain $\psi^{(t)}$. Let the flattened dimension of $\psi^{(t)}$ be $d$, then the Hessian $H \in \mathbb{R^{d\times d}}$. All the parameters for the feature extractor $\phi^*$ will **not** need to be considered. Then, we can calculate the Hessian for the last layer by taking the second-order derivative $H_{{\psi}^{(t)},w^{(t)}} = \nabla_\psi^2 \hat{\mathcal{R}}({\cal D}^\mathrm{tr-h}; w^{(t)},{\psi}^{(t)})$. We have also updated line 13 of Algorithm 1 accordingly to show that only $\psi^{(t)}$ is used. We have added this discussion to Section 4.2 (lines 301-303).

---

> ### Author Response · Authors · 2024-11-21
> **Author Response (Part 3)**
>
> **Q5:**
>
> > Section 5.1: The method GSR-HF (Hessian-free) is examined as an ablation study, but how about the ablation study of using the Hessian but not LLR? To compare with MAPLE, this ablation study is also desired. (Perhaps is it too costly?)
> >
>
> We added the ablation study and revised the paper accordingly. Please kindly refer to the **General Response**.
>
> In addition, we have reimplemented and filled the missing entries for MAPLE in Tables 1 and 7, with more details in response to Weakness 4, Reviewer xdjG.
>
> ---
>
> We hope that our clarifications and additional experiments have addressed your concerns and will further improve your opinion of our work.
>
> ---
>
> **References:**
>
> [1] Weixin Liang and James Zou. Metashift: A dataset of datasets for evaluating contextual distribution shifts and training conflicts. arXiv preprint arXiv:2202.06523, 2022.
>
> [2] Yuzhe Yang, Haoran Zhang, Dina Katabi, and Marzyeh Ghassemi. Change is hard: A closer look at subpopulation shift. In International Conference on Machine Learning, 2023.

---

> ### Comment · Reviewer_wcXj · 2024-11-22
>
> Thank you for kindly responding my questions. The paper looks that the motivation become clearer.

---

> > ### Author Response · Authors · 2024-11-25
> >
> > Dear Reviewer wcXj, thank you for the prompt response and detailed review. We would like to check if you have any additional questions beyond our rebuttal and we hope to address them in time during the discussion period.

---

### Author Response · Authors · 2024-11-21
**General Response**

We thank all reviewers for their valuable feedbacks. We have revised the main paper accordingly, with changes highlighted in **blue**.

As suggested by Reviewer wcXj and kehk, we first include an ablation study on **full-parameter retraining** vs. **last-layer retraining** with GSR at a **manageable** scale, then demonstrate the challenges of performing GSR with full-parameter retraining at a **larger** scale such as for the subpopulation shift benchmarks.

### **Comparing full-parameter retraining (FR) vs. last-layer retraining (LLR) with GSR**

Due to the computational constraints, we design a small binary classification task *ColoredMNIST-S*, customized from the ColoredMNIST dataset [1] with 10000 training examples, where the digits are spuriously correlated with color (90% of all 1s are green and 90% of all 0s are red), creating 4 groups. We use a tiny Convolutional Neural Network with 3 layers and about 5000 parameters. A more detailed description of the setup and dataset is available in the Appendix E. We compare the results of training with:

1. GSR-FR-EH: Full-parameter retraining (FR) on GSR with “exact” Hessian, calculated with Conjugate Gradients (CG) method.
2. GSR-FR-AH: FR on GSR  approximated Hessian (calculated with LiSSA [2], which approximates the Hessian with 10% of the training data),
3. GSR-FR-HF: FR with gradient inner product (i.e., MAPLE)
4. Our two-stage GSR (with LLR) and GSR-HF (i.e., MAPLE+LLR).

| **Approach** | Worst-group Accuracy | Total Wall-clock Time (min) |
| --- | --- | --- |
| GSR-FR-EH (Full-param retraining with “exact” Hessian (CG)) | **94.1** | 400 |
| GSR-FR-AH (Full-param retraining with approximated Hessian (LiSSA [2])) | 80.2 | 45 |
| GSR-FR-HF (Hessian-free Full-param retraining, i.e., MAPLE) | 91.9 | 42 |
| GSR (LLR with Hessian) | 91.2 | 4+0.8 |
| GSR-HF (LLR without Hessian, i.e., MAPLE+LLR) | 91.1 | 4+0.7 |
| ERM | 81.2 | 4 |

From the table, we can observe that at a smaller scale:

1. Full-parameter retraining with Hessian is advantageous over all other approaches, but at a much higher cost.
2. Approximating the Hessian with low quality might hurt the performance.
3. When using a small neural network model, LLR can cause slight performance degradation when we have the capability to retrain the whole model on the entire dataset, but it is still significantly better than ERM.

This result suggests that we should consider FR when we have the capacity, and LLR is an effective approximation to FR.

---

### **Challenges of scaling up the full-parameter retraining with GSR**

However, when scaling up full-parameter retraining to more realistic datasets with larger networks, **the computational challenges for accurately calculating the sample weight gradients are indeed the major obstacle here because of the Hessian.** Let $n$ be the number of data points and $p$ be the number of parameters. Directly calculating the Hessian for the whole network is both computationally expensive $O(np^2+p^3)$ and memory bottlenecked $O(p^2)$. In particular, the Hessian of a simple neural network with 1M parameters takes 4TB of memory, and ResNet-50 has 23M trainable parameters. Alternatively, we can calculate the inverse Hessian-vector product (iHVP) to reduce the memory requirements to $O(p)$ and the time complexity to $O(np^2)$, but it is still time-consuming, unless we apply further approximations. The table below shows the computational challenges of performing full-parameter tuning with GSR on the Waterbirds dataset:

| **Approach** | **Time (One-iteration)** | **VRAM Consumption** |
| --- | --- | --- |
| Explicit Hessian | >18h | 2PB |
| Conjugate Gradients | 18h | 9GB |
| LiSSA | 3.6h | 9GB |

It will be extremely costly to perform the experiments that optimizes the sample weights for 50 iterations as done with LLR. Besides, as shown in the previous ablation study, training the full network with crudely approximated Hessian (using LiSSA) causes inaccurate gradient estimation and may even hurt the performance. We believe these technical challenges require extensive exploration in future works and are beyond the scope of this paper, as originally discussed in Section 7 (lines 536-539),

[1] Martin Arjovsky, Léon Bottou, Ishaan Gulrajani, and David Lopez-Paz. Invariant risk minimization. arXiv preprint arXiv:1907.02893, 2019.

[2] Naman Agarwal, Brian Bullins, and Elad Hazan. Second-order stochastic optimization for machine learning in linear time. Journal of Machine Learning Research, 18(116):1–40, 2017.

---

### Author Response · Authors · 2024-12-04
**Rebuttal Summary**

We thank all reviewers for their precious time and feedback during the rebuttal. Below, we summarize the reviews and discussions:

- **Reviewer wcXj** found our approach effective and computationally efficient in tackling subpopulation shifts, but raised the following concerns:
    - *Presentation:* We have carefully revised the paper and incorporated additional diagrams.
    - *Empirical studies for additional datasets and cases where last-layer retraining might be insufficient:* we added experiments with additional datasets and demonstrated that our approach remains advantageous.
    - *Empirical study involving full-parameter retraining with Hessian:* we introduced an additional ablation study that exhaustively compares the baselines and variants at a manageable scale.
- **Reviewer kehk** recognized the technical contribution of our approach and the thoroughness of our experiments, but was concerned about:
    - *The relationship between last-layer retraining and other factors, such as our assumption and regularization:* We have revised the paper to include additional clarifications and detailed explanations.
    - *Comparison with full-parameter retraining:* We exhaustively compared baselines involving full-parameter and last-layer retraining at a manageable scale, highlighting the advantages of using the Hessian.
- **Reviewer xdjG** recognized our theoretical solidity and the comprehensiveness of the experiments, but raised the following concerns:
    - *Comparisons with MAPLE:* we have updated the main paper to clearly state our differences, amended the empirical results for MAPLE, and clarified that the GSR-HF baseline corresponds to MAPLE+LLR, as requested by the reviewer.
    - *Validity of a claim in Section 4:* we have revised the confusing claim and provided supporting evidence including references to relevant literature and additional experiments.
- **Reviewer oDPM** acknowledged the validity, effectiveness, and noise robustness of our approach, but was concerned about the *theoretical derivations*. We have addressed these concerns by incorporating further clarifications and explanations in the revision.

---

### Meta-Review · Area_Chair_So8i · 2024-12-19

**Metareview:**

This submission is a borderline case. It worked on subpopulation shift (a special case of distribution shift) with spurious correlations, and proposed group-robust sample reweighting (GSR) utilizing group-labeled data to optimize the weights of other group-unlabeled data, to effectively reduce the required amount of group labels. After the rebuttal, all four reviewers gave a rating of 6 that is "marginally above the acceptance threshold". During the internal discussions, none of four positive reviewers showed up, even after I cued them, suggesting that they were not truly supportive.

I quickly checked this paper and found that I shared exactly the same concerns with Reviewer xdjG. Subsequently, I found that the authors did a particularly good job of addressing his or her concerns (and thus mine). I like very much the first point in the rebuttal about the conceptual or philosophical difference between the proposed GSR and the baseline MAPLE, which emphasized the novelty of this work. The third point is also interesting about the motivation or justification of last-layer retraining in GSR. Since the motivation is strong and the contributions are solid (at least to me), I think we should accept this paper for publication even though our reviewers are only marginally positive.

I feel the authors considered themselves doing distribution shift research (they mentioned so in the rebuttal letters). There are some related papers you might be interested in about distribution shift and importance weighting for deep learning: "Rethinking importance weighting for deep learning under distribution shift", NeurIPS 2020 spotlight; "Generalizing importance weighting to a universal solver for distribution shift problems", NeurIPS 2023 spotlight. For certain hard distribution shift problems, last-layer retraining is not enough, and the papers tried to improve "the representations from a poor ERM-trained model" by iteratively updating all layers of the model and updating the weights on the data.

**Additional Comments On Reviewer Discussion:**

During the internal discussions, none of four positive reviewers (whose ratings were all 6) showed up, even after I cued them, suggesting that they were not truly supportive.

---

### Decision · Program_Chairs · 2025-01-22

Accept (Poster)